# Learning Transferable Interaction Primitives from Game Videos for Humanoid Locomotion

**Xiangming Zhu** [1]  **Huayu Deng** [1]  **Haoran Zhao** [1]  **Yiwei Hao** [1]  **Yunbo Wang** [1]

## Abstract

Learning humanoid control from video provides a scalable alternative to the scarcity of high-fidelity robot data. Existing methods, however, often rely on curated datasets and treat video as passive kinematic priors. They fail to capture dynamic humanoid interactions with the environment, which are essential for robust control in complex physical environments. To address this, we propose *TRansferable Interaction Primitives (TRIP)*, a framework designed to extract and ground interactions from unlabeled game videos for locomotion control. TRIP explicitly models dependencies between motion dynamics and environmental context via a discrete library of interaction-based action primitives. To bridge the reality gap, we introduce a shared context latent space that aligns implicit video-domain features with functional target-domain observations, enabling the seamless transfer of video-mined strategies to reinforcement learning policies. Our experiments on complex terrain navigation demonstrate that TRIP achieves significant improvements in task performance, sample efficiency, and robustness.

## 1. Introduction

Learning robotic control from readily available video data offers a promising and scalable paradigm for equipping robots with agile, human-like skills. By leveraging the rich behavioral diversity in videos, simulated robots can receive extensive supervision without requiring costly or potentially unsafe real-world trials (Pan et al., 2025; Li et al., 2024; Ni et al., 2025). However, most existing approaches prioritize passive kinematic imitation without explicitly modeling or disentangling the underlying *humanoid–environment inter-*

*Table 1.* **Comparison of Humanoid Control Methods.** Unlike prior approaches that directly mimic humanoid motions from videos, TRIP transfers interaction knowledge with a learned motion prior that captures the distribution of rich humanoid motions and to novel physical environments.

| Model | Learn from Video | Motion Prior | Interaction Modeling |
|---|---|---|---|
| PHC (2023) | ✗ | ✗ | ✗ |
| PULSE (2024a) | ✗ | ✓ | ✗ |
| OKAMI (2024) | ✓ | ✗ | ✗ |
| GenMimic (2025) | ✓ | ✗ | ✗ |
| **TRIP (Ours)** | ✓ | ✓ | ✓ |

*actions*, such as adapting gait to varying terrain and contact conditions, that govern such behaviors. As summarized in Table 1, motion mimicry (Luo et al., 2023) struggles to generalize beyond expert demonstrations, while motion-prior approaches (Luo et al., 2024a; Wang et al., 2024b) typically assume over-simplified, static environments.

Consider a humanoid traversing uneven terrain: success depends not on replaying a static sequence, but on continuous sensorimotor adaptation to local surface geometry and fluctuating contact forces (Radosavovic et al., 2024). This underscores a fundamental challenge: How to extract and transfer the **humanoid–environment interactions** implicit in passive videos and transfer it into control policies that are robust in more realistic settings. This challenge is further compounded by a critical technical bottleneck. Although environmental context is indispensable for grounding interaction, it is inherently unobserved and difficult to recover from monocular video alone.

To address these issues, we propose **TRIP**, an interaction-centric framework that learns reusable interactions from videos and grounds them for execution in novel physical environments. As illustrated in Figure 1, our key insight is to decouple *how* an agent physically interacts with its surrounding environment from *when* such interactions should be invoked. Accordingly, TRIP first extracts a library of discrete interaction primitives from human motion in videos. These primitives capture short-horizon motor strategies that reflect interaction patterns shaped by the environment's terrain geometry, yet remain deliberately ungrounded and independent of any explicit environmental representation.

[1]MoE Key Lab of Artificial Intelligence, Institute of AI, School of Computer Science, Shanghai Jiao Tong University. Correspondence to: Yunbo Wang <yunbow@sjtu.edu.cn>.

*Proceedings of the 43rd International Conference on Machine Learning*, Seoul, South Korea. PMLR 306, 2026. Copyright 2026 by the author(s).

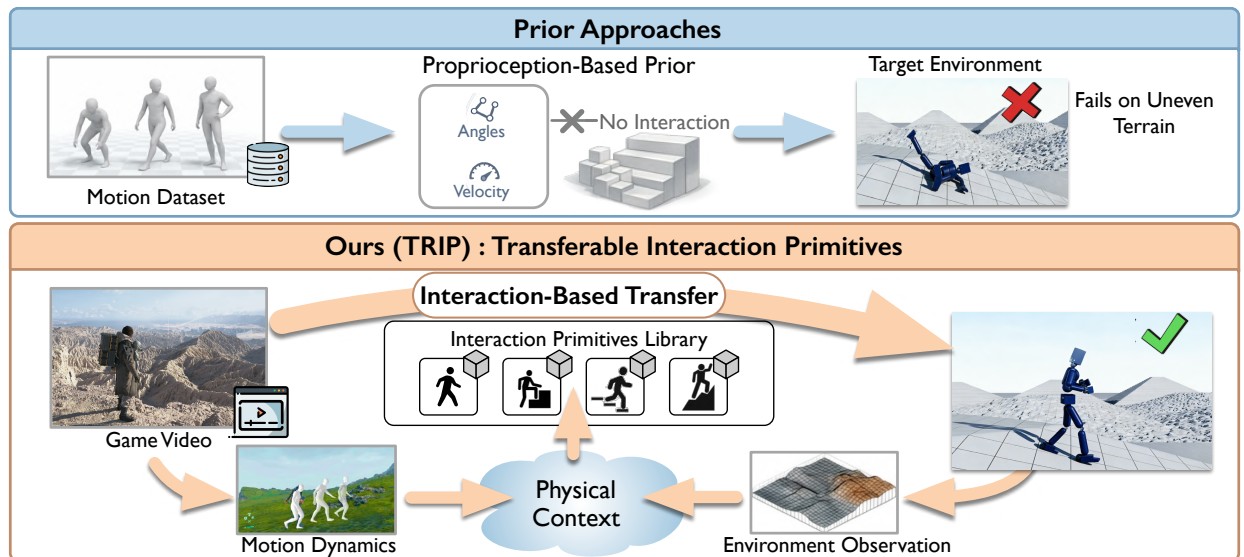

*Figure 1.* **Key insight.** While existing works focus on imitating kinematic motion priors from curated datasets, TRIP extracts interaction-based action primitives from unconstrained videos and grounds them in novel environments via a shared context latent space.

To enable real-world execution, we ground these primitives through a *shared, domain-invariant latent space* that captures the physical context. This latent variable is learned by aligning two complementary views: one inferred from environment-aware observations in the target domain, and the other inferred solely from motion dynamics in the video domain. By enforcing consistency between these latent representations across domains, TRIP establishes a cross-domain bridge that allows a policy to correctly select and compose video-learned interaction primitives in response to novel and dynamically changing environments.

The contributions of this work are summarized as follows:

- We propose TRIP, an interaction-centric framework for distilling *humanoid–environment interaction primitives* from passive, unlabeled videos and transferring them to physically executable control policies.

- We develop a cross-domain latent alignment approach that bridges video and physical domains via a shared, domain-invariant context representation, enabling seamless policy transfer without paired demonstrations.

- We show that executing the learned interaction primitives significantly enhances the robustness and generalization of humanoid policies across challenging terrains.

## 2. Problem Formulation

We aim to learn transferable humanoid–environment interactions from passive videos, such that the policies can perform closed-loop humanoid control in novel physical environments. A key challenge of this problem arises from the asymmetry between the video domain and the target robotic domain in both observability and interaction modeling.

- **The video domain** ($\mathcal{V}$). We assume access to a dataset of human motion videos (*e.g.*, gameplay), which contains rich human-environment interactions implicitly expressed by motion. Crucially, the video domain provides *only visual observations*, without access to underlying environment configurations that govern these interactions.

- **The physical domain** ($\mathcal{T}$). In the target physical domain, the humanoid agent operates in a physics environment with privileged access to proprioceptive states $s_t \in \mathcal{S}$ and explicit local environment and task observations $h_t \in \mathcal{H}$ at each time step $t$.

Our goal is to leverage interaction knowledge embedded in video data to learn a control policy $\pi_\theta(a_t \mid s_t, c_t)$ in the target physical domain, which is optimized to maximize the expected cumulative return $J(\pi) = \mathbb{E}_\pi \left[ \sum_t \gamma^t r(s_t, a_t) \right]$.

## 3. Preliminaries

Prior work learns compact humanoid motion priors from large-scale data. A representative is PULSE (Luo et al., 2024a), which learns from the AMASS dataset (Mahmood et al., 2019) by distilling a pretrained motion imitator into a variational latent space. Given the proprioceptive state $s_t$ and target state $s_t^g$, a variational encoder $\mathcal{E}(z_t|s_t, s_t^g)$ infers a latent embedding, which is decoded by $\mathcal{D}(a_t|s_t, z_t)$ to produce actions that track $s_t^g$. A proprioception prior $\mathcal{R}(z_t \mid s_t)$ regularizes the latent space by conditioning solely on $s_t$. The model is trained via online distillation from a pretrained tracking policy, using a reconstruction loss and a KL term to align the inferred posterior with the prior.

The learned latent space serves as a compact and well-regularized action representation, facilitating efficient down-

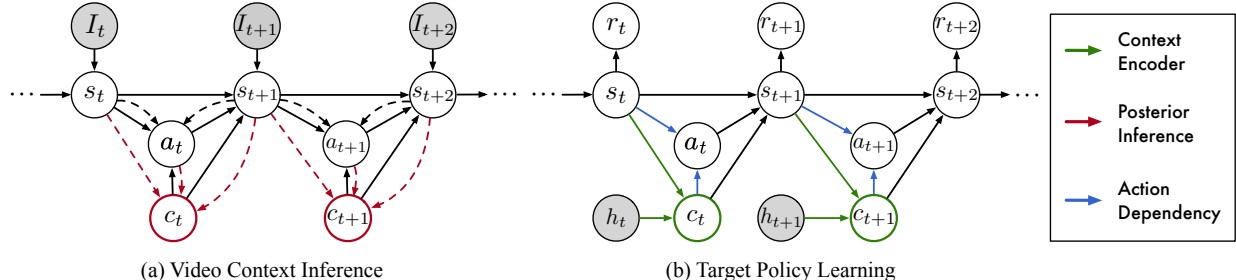

(a) Video Context Inference      (b) Target Policy Learning

*Figure 2.* **The TRIP graphical model.** (a) Posterior physical context inference from motion states and approximated actions in the video domain. (b) An interaction-conditioned primitive policy with the physical context grounded in environmental observations.

stream reinforcement learning (Luo et al., 2024b; Wu et al., 2025), where the prior $\mathcal{R}$ and decoder $\mathcal{D}$ are fixed. Task policy $\pi_{\text{task}}$ learns a residual mean offset relative to prior $\mathcal{R}(z_t|s_t)$. The latent $z_t$ is sampled as: $z_t \sim \mathcal{N}(\mu_t^{\text{task}} + \mu_t^p, \sigma_t^{\text{task}})$, where $\mu_t^{\text{task}}, \sigma_t^{\text{task}} = \pi_{\text{task}}(s_t, h_t)$. Here, $s_t$ denotes the proprioceptive state, $h_t$ represents the task-specific goal, and $\mu_t^p$ is the mean of the prior $\mathcal{R}(z_t|s_t)$. The latent space of $\pi_{\text{task}}$ is thus formed as a residual feature with respect to the proprioception-based prior, promoting structured exploration and injecting human-like motion priors into downstream learning. However, conditioning the prior solely on proprioceptive state can be inherently ambiguous, as the same internal state may correspond to multiple possible physical contexts and patterns.

## 4. Method

### 4.1. Overview

To bridge the gap between in-the-wild videos and humanoid control in the target domain, we propose **TRIP**, which reformulates the ill-posed problem of direct video-to-control transfer as structured interaction transfer through a shared, domain-invariant latent interface. We first learn a set of cross-domain interaction primitives from videos. Each primitive captures a short-horizon interaction pattern $g_{k_t}$ inferred solely from video motion dynamics (**Section 4.2**).

However, deploying these primitives in a novel physical environment is non-trivial. While primitives encode the evolution of motions, how and when these motions should interact the specific environment remains undefined. To physically ground interaction primitives, we introduce a domain-invariant representation $c_t \in \mathcal{C}$ for the environmental context to determine how primitives should be executed under different environmental conditions (**Section 4.3**). As illustrated in Figure 2, the latent context $c_t$ is inferred through two domain-specific pathways. In the video domain, the physical context is inferred from motion dynamics, using a video-context posterior model $q(c_t \mid s_t, a_t, s_{t+1})$, capturing implicit information about the underlying interaction regime. In the target robotic domain, $c_t$ is predicted via a context encoder $p(c_t \mid s_t, h_t)$ leveraging the robot's proprioceptive

state $s_t$ and environment observation $h_t$.

Based on the aligned physical context, we train a context-guided policy $\pi_\theta(a_t \mid s_t, c_t)$ via reinforcement learning (RL) in the target domain (**Section 4.4**). This design enables TRIP to leverage passive videos to acquire transferable policies, while exploiting privileged environment information in the target domain to correctly instantiate interaction behaviors during closed-loop deployment. The overall TRIP pipeline is illustrated in Figure 3.

### 4.2. Interaction Primitives Learning from Video

The video domain provides only visual sequences of the humanoid, with **no direct access** to detailed observations such as surrounding terrain heightmaps. We first reconstruct the humanoid's kinematic motion from raw gameplay footage. The resulting motion sequences, denoted as $\{s_{1:T}^{(i)}\}_{i=1}^N$, form the foundation of our primitive learning.

Learning control policies directly from reconstructed human pose sequences is insufficient, as such motions lack the environmental context that elicits the motion. Instead, we distill reusable **interaction primitives** from passive videos, forming a discrete codebook of short-horizon motion patterns that implicitly capture humanoid–environment interactions, while remaining independent of any explicit environment representation. The codebook is learned via unsupervised discretization of continuous motion representations with a VQ-VAE-based (van den Oord et al., 2017) framework.

The primitive learning process operates on sliding windows of motion states, extracted from the video domain. We leverage two pretrained components from PULSE (Luo et al., 2024a) to extract high-level motion semantics:

- Proprioceptive prior: A static prior $\tilde{z}_t \sim \mathcal{R}(\tilde{z}_t \mid s_t)$ that encodes the proprioceptive context of the initial state $s_t$.

- Kinematic latent sequence $z_{t:t+\tau}$: A latent sequence produced by the variational encoder $\mathcal{E}$, which captures the temporal evolution from $s_t$ to $s_{t+\tau+1}$, representing how the interaction unfolds over time[1].

---
[1]Specifically, $z_{t:t+\tau}$ is generated by iteratively applying $\mathcal{E}$ to consecutive state pairs $(s_k, s_{k+1})$ within the window.

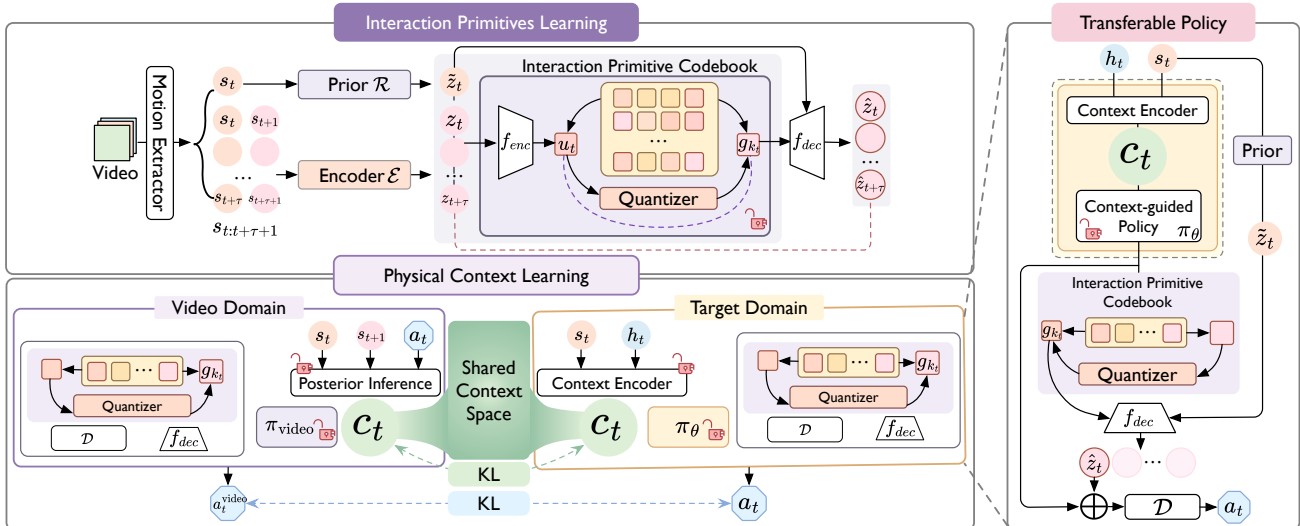

*Figure 3.* **TRIP framework.** *Top.* TRIP distills discrete interaction primitives from passive videos by encoding motion sequences within a temporal sliding window and quantizing them into a shared codebook. *Bottom.* TRIP aligns physical contexts inferred from video dynamics and target-domain environmental observations into a shared latent space. *Right.* Conditioned on the aligned physical context and learned interaction primitives, TRIP learns a context-guided policy that generalizes to novel physical environments.

We concatenate the static prior with the kinematic latent sequence and map them through an encoder $f_{\text{enc}}$ to a compact embedding $u_t = f_{\text{enc}}([\tilde{z}_t; z_{t:t+\tau}])$. This design enables the proposed interaction primitive to account for both the initial condition and the full kinematic trajectory of the interaction.

To obtain interpretable and reusable interaction primitives, we discretize the embedding space using vector quantization (VQ), mapping the continuous embedding $u_t$ to its nearest entry in a learnable codebook $\mathcal{G} = \{g_k\}_{k=1}^K$, yielding a discrete code $g_{k_t} = \text{argmin}_{g_k \in \mathcal{G}} \|u_t - g_k\|_2^2$. This quantization acts as an information bottleneck, compressing continuous motion dynamics into a finite set of interaction prototypes.

To capture full temporal semantics of interaction, a decoder $f_{\text{dec}}$ reconstructs the dynamic sequence $\hat{z}_{t:t+\tau} = f_{\text{dec}}(g_{k_t}, \tilde{z}_t)$ conditioned on the code $g_{k_t}$ and the initial static prior $\tilde{z}_t$. This conditioning enables each primitive $g_{k_t}$ to represent a relative motion strategy adaptive to the current proprioceptive context. The entire module is trained end-to-end with the following objective:

$$\mathcal{L}_{\text{prim}} = \underbrace{\|z_{t:t+\tau} - \hat{z}_{t:t+\tau}\|_2^2}_{\mathcal{L}_{\text{rec}}} + \underbrace{\|\text{sg}[u_t] - g_{k_t}\|_2^2}_{\mathcal{L}_{\text{codebook}}}$$
$$+ \underbrace{\beta \|u_t - \text{sg}[g_{k_t}]\|_2^2}_{\mathcal{L}_{\text{commitment}}}, \qquad (1)$$

where $\text{sg}[\cdot]$ denotes stop gradient.

Upon convergence, we obtain a fixed interaction primitive codebook $\mathcal{G}$. These primitives are ungrounded, possessing no knowledge of under what environmental conditions this pattern should be invoked. As composable building blocks, these interaction primitives provide a compact and reusable basis for complex locomotion behaviors.

### 4.3. Cross-Domain Physical Context Alignment

While interaction primitives provide a reusable motion vocabulary, they remain ungrounded: they encode the kinematic *how*, but lack the environmental *when*. This limitation is inherent to the video domain, where explicit environmental information (*e.g.*, precise geometry or contact forces) are unobservable. To bridge this "grounding gap", we must associate primitives with specific environmental contexts.

We introduce a shared latent variable, the **physical context** $c_t$, which represents the environment-dependent factors that trigger specific interaction patterns. To cope with asymmetric domain inputs, where video data provides only motion states, while the target domain exposes explicit environmental observations, the physical context $c_t$ is learned in a domain-aligned latent space, such that contexts inferred from video dynamics are semantically equivalent to those inferred from environmental observations during control. Specifically, we employ two domain-specific inference pathways that map heterogeneous inputs to $c_t$:

$$\text{Context encoder:} \quad c_t \sim p_\psi(c_t \mid h_t, s_t), \qquad (2)$$
$$\text{Posterior inference:} \quad \tilde{c}_t \sim q_\phi(\tilde{c}_t \mid s_t, a_t, s_{t+1}), \qquad (3)$$

where the context encoder uses explicit environment data $h_t$ to determine the context, while the Posterior Inference recovers the physical context by observing the resulting motion dynamics $(s_t, a_t, s_{t+1})$ with pseudo action labels inferred from a pretrained reference policy (Luo et al., 2023) in the video domain. To ensure $c_t$ captures physically meaningful and predictive factors, we regularize the latent space

by conditioning both the transition dynamics and the task reward on it:

Transition model: $\quad s_{t+1} \sim f_\xi(s_{t+1} \mid s_t, a_t, c_t)$, (4)

Reward model: $\quad r_t \sim f_r(r_t \mid s_t, c_t)$. (5)

This encourages $c_t$ to encode information about how the state evolves and whether the interaction succeeds. The overall objective of physical context learning is:

$$\mathcal{L}_{\text{ctx}} = \lambda_{\text{alg}} \underbrace{\mathcal{D}_{\text{KL}}(q_\phi(\tilde{c}_t \mid s_t, a_t, s_{t+1}) \,\|\, p_\psi(c_t \mid h_t, s_t))}_{\mathcal{L}_{\text{alignment}}}$$
$$+ \lambda_s \underbrace{\left( -\log p_\xi(s_{t+1} \mid s_t, a_t, c_t) \right)}_{\mathcal{L}_{\text{dynamics}}}$$
$$+ \lambda_r \underbrace{\left( -\log p_r(r_t \mid s_t, c_t) \right)}_{\mathcal{L}_{\text{reward}}}. \quad (6)$$

Minimizing $\mathcal{L}_{\text{alignment}}$ enables physical context inferred from *motion-only observations* in the video domain to be semantically consistent with *environment-aware physical context* in the target domain, thereby facilitating cross-domain transfer of interaction primitives.

### 4.4. Interaction Policy Learning

We develop an RL method to learn interaction policies conditioned on the domain-aligned physical context $c_t$, enabling the selection and adaptation of primitives for control. The policy adopts a hierarchical structure: it first selects a discrete primitive from the codebook $\mathcal{G}$ based on $c_t$, and subsequently applies a continuous residual for fine-grained execution. Formally, the policy $\pi_\theta$ maps the proprioceptive state $s_t$ and the physical context $c_t$ to two outputs:

$$(\ell_t, \delta_t) = \pi_\theta(s_t, c_t), \quad (7)$$

where $\ell_t \in \mathbb{R}^K$ are logits over the $K$ interaction primitives, and $\delta_t$ is a continuous residual vector for fine-grained adaptation. A primitive index $k_t$ is sampled from the categorical distribution defined by $\ell_t$. The corresponding primitive feature $g_{k_t} \in \mathcal{G}$, and the state-dependent prior $\tilde{z}_t \sim \mathcal{R}(z_t \mid s_t)$ (from Section 4.2), is fed into the pretrained decoder $f_{\text{dec}}$ to reconstruct a short-horizon latent trajectory: $\hat{z}_{t:t+\tau} = f_{\text{dec}}(g_{k_t}, \tilde{z}_t)$. We adopt a receding-horizon execution scheme, where only the first latent variable $\hat{z}_t$ is utilized at the current time step. The control action $a_t$ is produced by a pretrained action decoder $\mathcal{D}$ (*e.g.*, from PULSE (Luo et al., 2024a)):

$$a_t = \mathcal{D}(\hat{z}_t + \delta_t). \quad (8)$$

The residual $\delta_t$ allows the policy to make continuous adjustments to the nominal motion pattern in response to local environmental nuances.

To leverage behavior structure from video, we use two policies with identical architectures: $\pi_{\text{video}}$ for the video domain

and $\pi_\theta$ for the target RL domain. $\pi_{\text{video}}$ is trained via behavior cloning on reconstructed motion using the retrieved primitives and pseudo actions, while $\pi_\theta$ is optimized with RL to maximize task rewards. We alternate between imitation updates for $\pi_{\text{video}}$ and RL updates for $\pi_\theta$, ensuring the control policy remains grounded in both demonstrated interactions and task rewards. We introduce a value-difference-weighted regularizer that encourages $\pi_\theta$ to align with $\pi_{\text{video}}$ specifically when the video-derived policy is predicted to yield higher returns:

$$\Delta_t = \text{clip}(\hat{Q}_t^{\pi_{\text{video}}} - \hat{Q}_t^{\pi_\theta}, -\epsilon, \epsilon). \quad (9)$$

where $\hat{Q}_t$ is estimated using the learned transition and reward models:

$$\hat{Q}_t^\pi = \hat{r}_t^\pi + \gamma \hat{V}(\hat{s}_{t+1}^\pi). \quad (10)$$

The regularization loss is defined as:

$$\mathcal{L}_{\text{reg}} = \mathbb{E}\Big[\Delta_t \cdot \mathcal{D}_{\text{KL}}\big(\pi_\theta(\cdot|s_t, c_t) \,\|\, \text{sg}[\pi_{\text{video}}(\cdot|s_t, c_t)]\big)$$
$$+ \lambda \, \mathcal{D}_{\text{KL}}\big(\pi_{\text{video}}(\cdot|s_t, c_t) \,\|\, \text{sg}[\pi_\theta(\cdot|s_t, c_t)]\big)\Big]. \quad (11)$$

This regularization ensures that $\pi_\theta$ selectively inherits expert knowledge from the video domain when beneficial, while the reverse term prevents the two policies from diverging during training, maintaining stable exploration.

## 5. Experiments

### 5.1. Experimental Setup

**Video dataset curation.** We construct the video-domain training data from gameplay recordings of *Death Stranding*[2], which feature a diverse range of interaction patterns. Gameplay sequences are recorded as the character traverses complex terrains, including slopes, gravel roads, and uneven ground. All footage is captured at 30 fps and segmented into 116 clips of 10 seconds each, forming the video dataset used in our experiments. We extract estimated motions as proprioceptive states $s_t$ from gameplay footage with pose estimator TRAM (Wang et al., 2024a). The derived motion sequences can be used for training both our method and the baselines. We detail motion reconstruction in Appendix C.

**Benchmarks.** We evaluate the TRIP on three target domains involving complex environment interactions: *(i)* **Trajectory**: the agent is required to track a randomly generated trajectory with target velocities sampled from $[0, 3]$ m/s and accelerations from $[0, 2]$ m/s²; *(ii)* **Speed**: the agent aims to achieve a target speed along the x-direction, sampled uniformly from $[0, 5]$ m/s; *(iii)* **Reach**: the agent is tasked with

---

[2]A video game by Kojima Productions https://www.kojimaproductions.jp/en/death-stranding-ps4

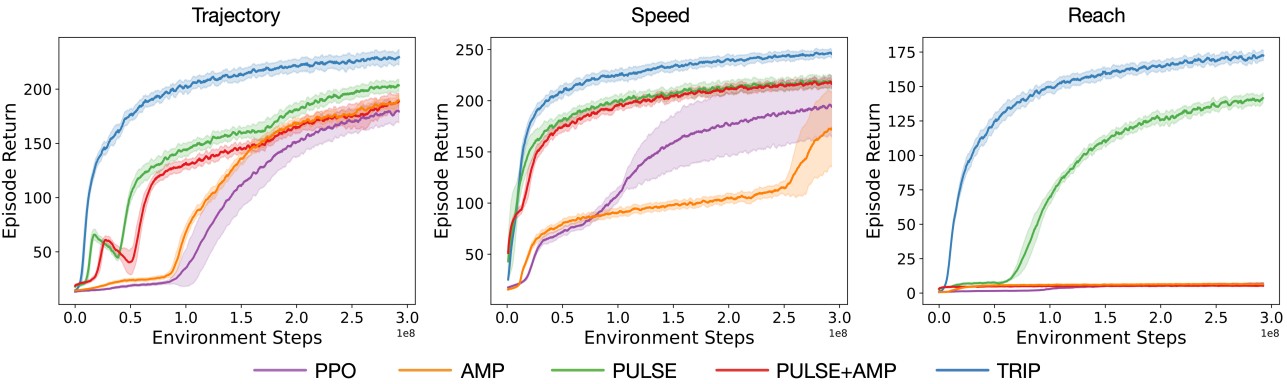

*Figure 4.* **Training curves for target physical deployment.** Leveraging learned interaction primitives and cross-domain physical context, TRIP achieves faster convergence and improved final performance by transferring rich interaction knowledge from the video domain to novel physical environments. Results are averaged over three runs with three random seeds.

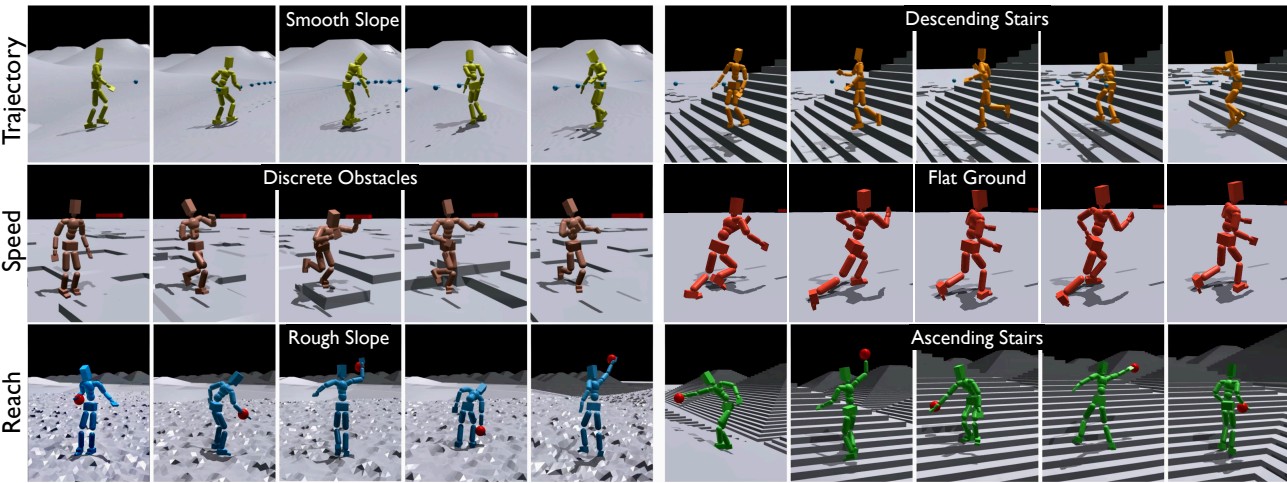

*Figure 5.* **Showcases of the behavior of TRIP interaction-conditioned policy across tasks.** Rollouts on three representative tasks—*Trajectory*, *Reach*, and *Speed*. TRIP transfers learned interaction primitives and cross-domain physical context to diverse and challenging terrain conditions.

reaching a target 3D point, where the target is initialized within a 2-meter radius of the humanoid. To evaluate the policy under complex physical conditions, we augment all tasks with terrain-rich environments using the IsaacGym framework (Makoviychuk et al., 2021), in contrast to prior works (Luo et al., 2024a; Peng et al., 2022) that primarily consider flat ground. Specifically, environments are embedded with terrains include *smooth slopes*, *rough slopes*, *staircases* (ascending and descending), *discrete obstacle ground*, and *flat ground*. Such environments induce diverse contact interactions and locomotion challenges. Task and environment details are provided in Appendix B.

**Compared methods.** We mainly compare TRIP with the following methods:

- PPO (Schulman et al., 2017), which is a reinforcement learning method trained from scratch on the target domain.
- PULSE (Luo et al., 2024a), which leverages a proprioceptive state-based prior learned from motion datasets.

- AMP (Peng et al., 2021), which trains a policy with an adversarial discriminator to encourage expert-like motions.
- PULSE + AMP (Luo et al., 2024b), which combines the proprioceptive prior with adversarial motion prior.

**Implementation details.** All experiments are conducted in IsaacGym (Makoviychuk et al., 2021). The policy runs at 30 Hz, while the simulator steps at 60 Hz. All modules are implemented as multi-layer perceptrons. The primitive codebook contains $K = 256$ entries with an embedding dimension of 128. All models are trained for $3 \times 10^8$ environment steps across tasks. Additional implementation details are provided in Appendix F.

### 5.2. Results

As shown in Figure 4, TRIP consistently achieves higher final performance and faster convergence than all baselines across tasks. Although AMP-based methods encourage realistic motions, they do not model humanoid–environment interactions and thus provide limited guidance for task-

*Table 2.* **Performance comparison under increased terrain difficulty.** Evaluation results on three tasks with $1.5\times$ scaled terrain parameters. TRIP achieves average promotion of 17.1%.

| Model ↓ | Trajectory | Speed | Reach |
|---|---|---|---|
| AMP (2021) | $153_{\pm 2.9}$ | $157_{\pm 2.9}$ | $5_{\pm 0.6}$ |
| PULSE (2024a) | $172_{\pm 14.2}$ | $196_{\pm 6.8}$ | $97_{\pm 3.7}$ |
| PULSE+AMP (2024b) | $161_{\pm 14.2}$ | $180_{\pm 2.8}$ | $5_{\pm 1.5}$ |
| TRIP | $\mathbf{197}_{\pm 3.6}$ | $\mathbf{216}_{\pm 3.4}$ | $\mathbf{123}_{\pm 8.2}$ |

*Table 3.* **Performance on unseen *wave-shaped* terrain.** TRIP outperforms baselines on tasks with unseen terrain geometry.

| Model ↓ | Trajectory | Speed | Reach |
|---|---|---|---|
| AMP (2021) | $160_{\pm 9.8}$ | $134_{\pm 22.1}$ | $6_{\pm 1.9}$ |
| PULSE (2024a) | $180_{\pm 7.6}$ | $214_{\pm 8.0}$ | $114_{\pm 2.3}$ |
| PULSE+AMP (2024b) | $173_{\pm 6.8}$ | $210_{\pm 10.0}$ | $5_{\pm 0.8}$ |
| TRIP | $\mathbf{223}_{\pm 11.9}$ | $\mathbf{232}_{\pm 7.2}$ | $\mathbf{183}_{\pm 1.8}$ |

*Table 4.* **Model Robustness to external disturbances during inference.** TRIP demonstrates stronger robustness and recovery capability enabled by interaction modeling.

| Model ↓ | Trajectory | Speed | Reach |
|---|---|---|---|
| AMP (2021) | $165_{\pm 8.4}$ | $169_{\pm 7.5}$ | $6_{\pm 0.8}$ |
| PULSE (2024a) | $176_{\pm 7.3}$ | $190_{\pm 6.9}$ | $134_{\pm 4.3}$ |
| PULSE+AMP (2024b) | $170_{\pm 5.7}$ | $191_{\pm 8.6}$ | $5_{\pm 0.7}$ |
| TRIP | $\mathbf{203}_{\pm 8.3}$ | $\mathbf{206}_{\pm 7.2}$ | $\mathbf{157}_{\pm 5.9}$ |

relevant behaviors. PULSE learns a prior representation from motions, but its static, proprioception-only conditioning ignores environmental context, leading to ambiguous latents and degraded control in novel environments with complex physical interactions. In contrast, TRIP explicitly models interaction dynamics by grounding motion in environmental physical context with interaction primitives learned from videos. By jointly reasoning over motion and environment, TRIP avoids the ambiguity of state-only priors and enables more effective adaptation to complex physical environments. This advantage is particularly evident in the Reach task, where the agent must maintain balance on irregular terrains while simultaneously tracking and reaching a target, posing a strong challenge to locomotion stability and balance control. We further present qualitative showcases of TRIP on three tasks across different terrain conditions in Figure 5. TRIP demonstrates strong balance recovery capabilities and human-like behaviors.

### 5.3. Generalization to Physical Variations

We evaluate the generalization ability of our approach under out-of-distribution physical variations not observed during training. Specifically, we systematically stress-test agents along three dimensions that introduce new challenges to the environment: increased terrain difficulty, unseen terrain geometries, and unexpected external disturbances, covering both structural changes and transient perturbations.

**Generalization to increased terrain difficulty.** We evaluate the trained policy on more challenging terrains than those seen during training by scaling slope angles, staircase step heights, and obstacle sizes by $1.5\times$ across tasks, while keeping the training distribution unchanged to assess generalization. As shown in Table 2, TRIP consistently demonstrates enhanced robustness and generalization, achieving an average performance gain of 17.6% over baselines. This advantage stems from leveraging rich interaction patterns from video and aligning physical context spaces across asymmetric domains, resulting in a more robust policy that maintains stable interactions under challenging terrain variations.

**Generalization to unseen terrain geometry.** We evaluate the generalizability of the learned policy on environments with terrain geometries that are entirely unseen during training. Specifically, we design a class of *wave-shaped terrains* characterized by periodic height variations, introduc-

ing rapid local gradient changes and continuous re-balancing demands that are absent from the training distribution. As shown in Table 3, TRIP outperforms baselines on unseen wave terrains. The results indicate that the learned policy does not overfit to specific terrain templates, but instead captures transferable physical interaction strategies and can generalize to structurally novel surface profiles.

**Robustness to external disturbances.** We evaluate robustness to unexpected external disturbances by randomly applying impulsive forces of fixed magnitude and random timing during execution, simulating sudden pushes or impacts. All policies are trained without such forces, enabling a direct assessment of robustness to unseen external disturbances. As shown in Table 4, TRIP consistently outperforms baseline methods across all tasks under external disturbances, achieving higher success rates and cumulative rewards. This demonstrates that TRIP learns more robust interaction and control strategies, allowing it to recover more effectively from sudden perturbations and maintain task performance under non-ideal execution conditions.

### 5.4. Model Analyses

**Ablation studies.** We conduct ablation studies on several variants of TRIP, including: *(i)* removing interaction primitives, *(ii)* disabling cross-domain context alignment (*i.e.*, removing $\mathcal{L}_{\text{alignment}}$), *(iii)* removing the regularization Eq. 11, and *(iv)* replacing Eq. 11 with one-sided regularization that only constrains $\pi_\theta$ to $\pi_{\text{video}}$. Figure 6 reports tracking error and task success rate on trajectory-following tasks. Removing interaction primitives ("w/o primitives") leads to a large drop in success rate, as the alignment module has no meaningful primitive vocabulary to select from, forcing the policy to learn interactions from scratch. Disabling cross-domain alignment ("w/o alignment") also degrades performance: the policy retains the structured primitive vocabulary and can partially recover primitive-environment associations through RL exploration, yet loses the systematic grounding

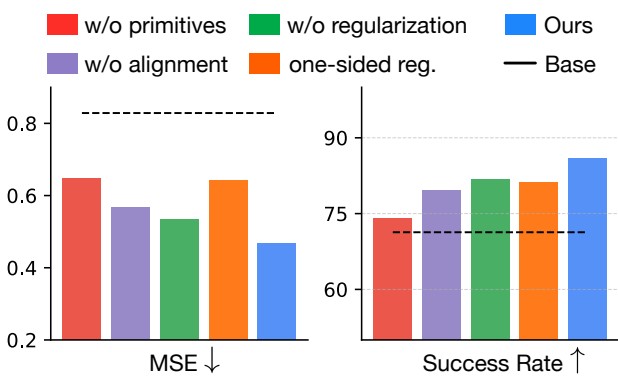

*Figure 6.* **Ablation studies on Trajectory task.** We provide model ablation of removing interaction primitives, $\mathcal{L}_{\text{align}}$, $\mathcal{L}_{\text{reg}}$, and using one-sided regularization in Eq. 11. The black dashed lines represent the results of PULSE.

*Table 5.* **Ablation on codebook size $K$.** $K=256$ achieves the best balance between reconstruction quality and codebook utilization.

| $K$ | Recon. Error | Dead Codes | Perplexity |
|---|---|---|---|
| 64 | 0.397 | 4 | 36.4 |
| 128 | 0.308 | 5 | 87.2 |
| 256 | 0.265 | 4 | 203.7 |
| 512 | 0.255 | 52 | 369.9 |

that transfers video-derived strategies to novel terrains. The regularization in Eq. 11 is designed to balance guidance and adaptability: the video policy provides a stable interaction prior for RL training, while mutual alignment prevents misalignment bias when the two domains differ. Consistent with this design, both removing the regularization ("w/o regularization") and enforcing a one-sided constraint ("one-sided reg.") result in inferior performance.

**Analyses of codebook size.** To validate the choice of codebook size, we ablate the codebook size $K \in \{64, 128, 256, 512\}$ and report reconstruction error ($\mathcal{L}_{\text{rec}}$ in Eq. 1), the number of dead codes, and codebook perplexity $\mathcal{H}$, which measures the effective number of utilized codes. As shown in Table 5, reconstruction error drops substantially from $K=64$ to $K=256$ but nearly saturates beyond this point. Meanwhile, $K=512$ produces 52 dead codes, indicating that the data does not support that level of granularity. At $K=256$, dead codes remain minimal and the effective usage rate reaches $79.6\%$, confirming that the codebook is well-utilized.

**Analyses of interaction primitives.** To better understand what the discrete primitive codebook captures, we visualize the codes $g_{k_t}$ selected by inputs from both the video domain and the target control domain. As shown in Figure 7, identical code indices are consistently associated with semantically coherent interaction regimes across domains. This qualitative alignment indicates that TRIP's codebook captures transferable interaction primitives within a shared physical context space, enabling consistent primitive selection and transfer despite domain differences.

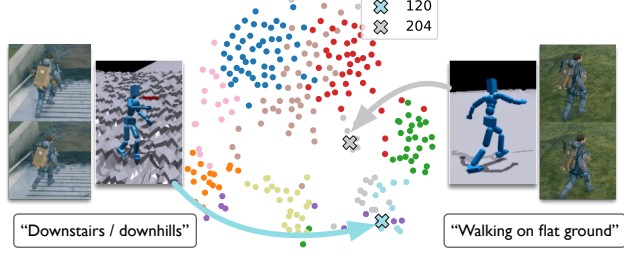

*Figure 7.* **Domain-aligned selection in the interaction primitive codebook.** We visualize the learned codebook in a 2D t-SNE space, with clusters obtained by K-Means. Motions are semantically mapped to the same interaction primitives across two domains.

*Table 6.* **Physical context transferability.** The posterior infers meaningful physical context from video-domain motion, significantly outperforming a random context baseline on next-state and action prediction.

| Context | State MSE ($\times 10^{-3}$) ↓ | Action MSE ($\times 10^{-3}$) ↓ |
|---|---|---|
| Random $\tilde{c}_t$ | 8.18 | 0.869 |
| Posterior $c_t$ | **4.48** | **0.581** |
| Reduction | 45.2% | 33.2% |

**Analyses of physical context.** We evaluate whether the posterior $q_\phi$ captures underlying physical structure that generalizes across domains by examining if the inferred context $\tilde{c}_t$ is sufficient for understanding motion in video sequences. Specifically, we freeze $q_\phi$ and train a lightweight predictor to estimate the next state $s_{t+1}$ and action $a_t$ solely from inferred context $\tilde{c}_t$. As a control, we replace $\tilde{c}_t$ with random vectors of the same dimension. As shown in Table 6, using $q_\phi$ reduces state and action prediction errors by $45.2\%$ and $33.2\%$, respectively, compared to the random baseline. This demonstrates that the inferred context encodes meaningful, transferable physical information.

**Generalization across video sources.** A natural question is whether the framework can leverage other video sources. The key requirement for our framework is interaction diversity induced by terrain variation, which modern text-to-video models can readily provide by generating diverse terrain traversal sequences at scale without any real-world data collection. To validate this, we train TRIP on 100 clips generated by text-to-video models (Seedance et al., 2026; Team et al., 2025) covering a wide range of terrain and motion patterns. The sampled frames are shown in Figure 9. As shown in Table 7, TRIP[†] consistently outperforms the baseline across all tasks, demonstrating that the framework can leverage scalable, readily obtainable video sources.

*Table 7.* **Generalization across video sources.** TRIP[†] denotes TRIP trained on AI-generated videos. Consistent gains demonstrate that the framework can learn from diverse source videos.

| Model | *Trajectory* | *Speed* | *Reach* |
|---|---|---|---|
| PULSE (2024a) | 212 | 238 | 147 |
| TRIP[†] | **232** | **243** | **170** |

## 6. Related Work

**Humanoid motion prior learning.** Motion priors aim to regularize humanoid control with human-like dynamics by leveraging large collections of reference motions (Haarnoja et al., 2018; Peng et al., 2021; 2022; Bae et al., 2023; Xu et al., 2023). A common strategy adopts adversarial objectives, where a discriminator encourages synthesized behaviors to match the distribution of real human motions (Peng et al., 2022; Dou et al., 2023; Tessler et al., 2023). SMP (Mu et al., 2025) replaces the adversarial discriminator with Score Distillation Sampling (Poole et al., 2023) to provide style-related rewards, serving as a reusable prior. Other approaches learn reusable motor representations through motion imitation and subsequently adapt them to downstream tasks (Yao et al., 2022; Zhu et al., 2023; Won et al., 2022). More recently, large-scale datasets (Mahmood et al., 2019) have enabled learning compact latent action spaces or distilled tracking policies (Luo et al., 2024a; Yao et al., 2024). Despite producing natural and stable motions, these priors are typically trained in relatively static settings, regardless of rich interactions with dynamically changing environments.

**Humanoid policy transfer from videos.** Learning robot control directly from videos provides a scalable alternative to reinforcement learning that avoids costly environment interaction (Torabi et al., 2018; Chang et al., 2022; Baker et al., 2022; Schmeckpeper et al., 2021; Ye et al., 2022; Pan et al., 2025; Chen et al., 2025). Early works aim to learn visual or action representation from video data (Bahl et al., 2022; 2023; Ma et al., 2022; Nair et al., 2022; Xiao et al., 2022; Radosavovic et al., 2023b;a). Benefiting from recent advances in human motion reconstruction (Wang et al., 2024a; Shin et al., 2024; Moon et al., 2024) and retargeting (He et al., 2024; 2025b; Tessler et al., 2024; Araujo et al., 2025), prior works enable motion replay from individual demonstrations (Ni et al., 2025; He et al., 2025a), faithfully reproducing reference behaviors but struggling to generalize to novel tasks. To tackle learning object manipulation from videos, a line of work explicitly incorporates object-centric modeling into video-based learning (Li et al., 2024; Weng et al., 2025; Wang et al., 2024b). However, they rely on correspondence between demonstrations and executions or require external object state estimation. SkillMimic (Wang et al., 2024b) builds a library of reusable low-level skills combined through hierarchical control, while the skills must be manually predefined. In contrast, our approach directly models humanoid-environment interactions and learns interaction primitives in latent space that generalize across tasks without paired demonstrations or manually designed skills.

## 7. Conclusions and Limitations

We introduced TRIP, an interaction-centric framework that learns transferable interaction primitives from unlabeled game videos and grounds them for humanoid control. By distilling a discrete library of interaction primitives and grounding them through a shared, domain-invariant physical context, TRIP establishes a structured bridge between motion dynamics in the video domain and environment-aware decision making in the target domain.

This work has several limitations. First, our method relies on video motion reconstruction, and reconstruction errors may affect subsequent learning. Second, the current formulation focuses on terrain-centric locomotion, while extending it to broader interaction categories, such as object manipulation and tool use, remains an important direction. Finally, applying the framework to real-world settings requires practical perception modalities, such as depth sensing or egocentric vision, as well as further study of sim-to-real transfer.

## Impact Statement

This work studies the learning of transferable interaction primitives from videos and their grounding for humanoid control. The proposed framework has the potential to reduce the cost and expertise needed to develop robust locomotion and interaction skills for robots. Such capabilities may benefit applications in assistive robotics, search and rescue, and operation in unstructured environments where manual policy design or large-scale data collection is challenging.

**Data sourcing and copyright.** The game-derived video data used in this work was captured from a legally purchased copy of Death Stranding (Kojima Productions, 2019) using non-invasive screen recording tools during normal gameplay, without reverse engineering or code modification. We do not use, store, or redistribute any original assets (e.g., textures, 3D models, audio, or source code), which is consistent with recent game-derived research datasets. Our use is strictly non-commercial, aligning with the game publisher's terms of use. All intellectual property rights pertaining to the original game assets remain with the game producers.

**Potential risks and harmful applications.** Our work is developed in controlled settings for terrain-centric locomotion and is solely for academic research and civilian use. We acknowledge potential misuse in safety-critical contexts. To mitigate these risks, we advocate restricting deployment to civilian domains, incorporating human oversight, and enforcing safety evaluation protocols. We strongly condemn any misuse of our work beyond its intended scope.

**Bias, safety, and responsible deployment.** Transferring behaviors from unconstrained videos to physical agents raises safety concerns, as models may inherit biases or unsafe behaviors. We mitigate this by manually curating the video data to retain only purposeful terrain traversal. We recommend behavioral auditing, safety compliance checks, and human-in-the-loop oversight before real-world deployment.

## Acknowledgements

This work was supported by the National Natural Science Foundation of China (62250062), the Smart Grid National Science and Technology Major Project (2024ZD0801200), the Shanghai Municipal Science and Technology Major Project (2021SHZDZX0102), Shanghai Jiao Tong University AI for Engineering Initiative (WH410263001/005), and the Fundamental Research Funds for the Central Universities.

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

# A. Algorithm

We present the full algorithm of TRIP in Alg. 1.

---

**Algorithm 1** Training pipeline of TRIP.

---

// Interaction primitives learning from video
Pretrain primitive codebook $\mathcal{G} = \{g_k\}_{k=1}^K$, encoder $f_{\text{enc}}$, decoder $f_{\text{dec}}$ following Sec. 4.2

---

// Policy learning with context alignment
Initialize target policy $\pi_\theta$, video policy $\pi_{\text{video}}$, context encoder $p_\psi(c_t \,|\, h_t, s_t)$, posterior $q_\phi(c_t \,|\, s_t, a_t, s_{t+1})$,
  transition model $f_\xi(s_{t+1} \,|\, s_t, a_t, c_t)$, reward model $f_r(r_t \,|\, s_t, c_t)$.
**while** *not converged* **do**
    // (1) Rollout collection in target domain
    Initialize buffer $\mathcal{D} \leftarrow \emptyset$.
    **for** *episode* $e = 1..E$ **do**
        **for** *time step* $t = 1 \to T$ **do**
            Infer physical context $c_t \sim p_\psi(c_t \,|\, h_t, s_t)$.
            Compute $(\ell_t, \delta_t) = \pi_\theta(\cdot \,|\, s_t, c_t)$ and sample $k_t \sim \text{Categorical}(\ell_t)$.
            Compute static prior $\tilde{z}_t \sim \mathcal{R}(\tilde{z}_t \,|\, s_t)$.
            Decode latent plan $\hat{z}_{t:t+\tau-1} = f_{\text{dec}}(g_{k_t}, \tilde{z}_t)$.
            Execute action $a_t \leftarrow \mathcal{D}(\hat{z}_t + \delta_t)$.
            Step env $\to (r_t, s_{t+1})$ and store $(s_t, h_t, c_t, k_t, \delta_t, a_t, r_t, s_{t+1})$ in $\mathcal{D}$.

    // (2) Context alignment update
    **for** *gradient step* $j = 1..N_{\text{ctx}}$ **do**
        Sample mini-batch $\mathcal{B} \sim \mathcal{D}$.
        Sample posterior $\tilde{c}_t \sim q_\phi(\tilde{c}_t \,|\, s_t, a_t, s_{t+1})$ and prior $c_t^{\text{prior}} \sim p_\psi(c_t \,|\, h_t, s_t)$.
        Minimize Eq. (6) to update $(\psi, \phi, \xi, r)$.

    // (3) Video-policy imitation update
    **for** *gradient step* $j = 1..N_{\text{BC}}$ **do**
        Sample video window $s_{t:t+\tau+1}$.
        Infer $c_t \sim q_\phi(c_t \,|\, s_t, a_t, s_{t+1})$.
        Update $\pi_{\text{video}}$ by behavior cloning on the motion sequence.

    // (4) Target policy optimization with distillation regularizer
    **for** *epoch* $e = 1..N_{\text{epoch}}$ **do**
        Sample mini-batch $\mathcal{B} \sim \mathcal{D}$.
        Compute PPO loss $\mathcal{L}_{\text{PPO}}(\theta)$.
        // Value-difference--weighted bidirectional regularization
        For each $(s_t, h_t) \in \mathcal{B}$, infer $c_t \sim p_\psi(c_t \,|\, h_t, s_t)$.
        Compute one-step estimates $\hat{Q}_t^\pi = \hat{r}_t^\pi + \gamma \hat{V}(\hat{s}_{t+1}^\pi)$ for $\pi \in \{\pi_{\text{video}}, \pi_\theta\}$.
        Compute $\mathcal{L}_{\text{reg}}$ via Eq. (11).
        Update $\theta \leftarrow \theta - \eta \nabla_\theta \big( \mathcal{L}_{\text{PPO}} + \alpha \mathcal{L}_{\text{reg}} \big)$.

---

# B. Benchmarks

### B.1. Task Description

We build three benchmarks with complex terrains, based on tasks defined in prior works. The speed and reach tasks follow the definitions in ASE (Peng et al., 2022), while the trajectory-following task is defined following PACER (Rempe et al., 2023). The environment observation $h_t$ is the concatenation of the terrain heightmap and task inputs. The terrain observation is a rasterized local height map of size $\mathbb{R}^{32 \times 32 \times 3}$, which captures a $2 \times 2 \ m^2$ square centered at the humanoid.

We define the full body human pose as $\mathbf{q}_t \triangleq (\boldsymbol{\theta}_t, \mathbf{p}_t)$, consisting of 3D joint rotation $\boldsymbol{\theta}_t \in \mathbb{R}^{J \times 6}$ and position $\mathbf{p}_t \in \mathbb{R}^{J \times 3}$ of all $J$ links on humanoid. To describe the movement of human motion, we include velocity $\dot{\mathbf{q}}_{1:T}$, where $\dot{\mathbf{q}}_t \triangleq (\boldsymbol{\omega}_t, \boldsymbol{v}_t)$ consists of angular $\boldsymbol{\omega}_t \in \mathbb{R}^{J \times 3}$ and linear velocities $\boldsymbol{v}_t \in \mathbb{R}^{J \times 3}$. We describe each task as follows:

- Trajectory. The trajectory-following task involves controlling a humanoid to follow random trajectories through stairs,

slopes, uneven surfaces (Rudin et al., 2022), and to avoid obstacles. The task input is defined as $x_{t:t+10}$, which is the next 10 time-step's 2D trajectory to follow. The reward is computed as $r_t^{\text{traj}} = \exp\left(-2\|\mathbf{p}_t^{(0)} - \boldsymbol{\tau}_t\|\right) - 0.0005 \cdot \sum_{j \in \text{joints}} |\boldsymbol{\mu}_j \dot{\mathbf{q}}_j|^2$ where the first item is the trajectory following the reward and the second term an energy penalty.

- Speed. For training the x-directional speed task, the random speed target is sampled between 0m/s $\sim$ 5m/s. The task input is defined as $(d_t, v_t)$ where $d_t$ is the target direction and $v_t$ is the linear velocity the policy should achieve at timestep t. The speed reward is defined as $r_{\text{speed}} = \exp\left(-\alpha\left((v_t - \mathbf{v}_t^{\parallel})^2 + \beta(\mathbf{v}_t^{\perp})^2\right)\right)$, where the root velocity is computed as $\mathbf{v}_t = \frac{\mathbf{p}_t - \mathbf{p}_{t-1}}{\Delta t}$.

- Reach. For the reach task, a 3D point $x_t$ is sampled from a 2-meter box centered at $(0, 0, 1)$. The task input is thus defined as the location of the target. The reward for reaching is the difference between the humanoid's right hand and the desired point position $r_{\text{reach}} = \exp(-5\|\mathbf{p}_t^{\text{right hand}} - x_t\|_2^2)$.

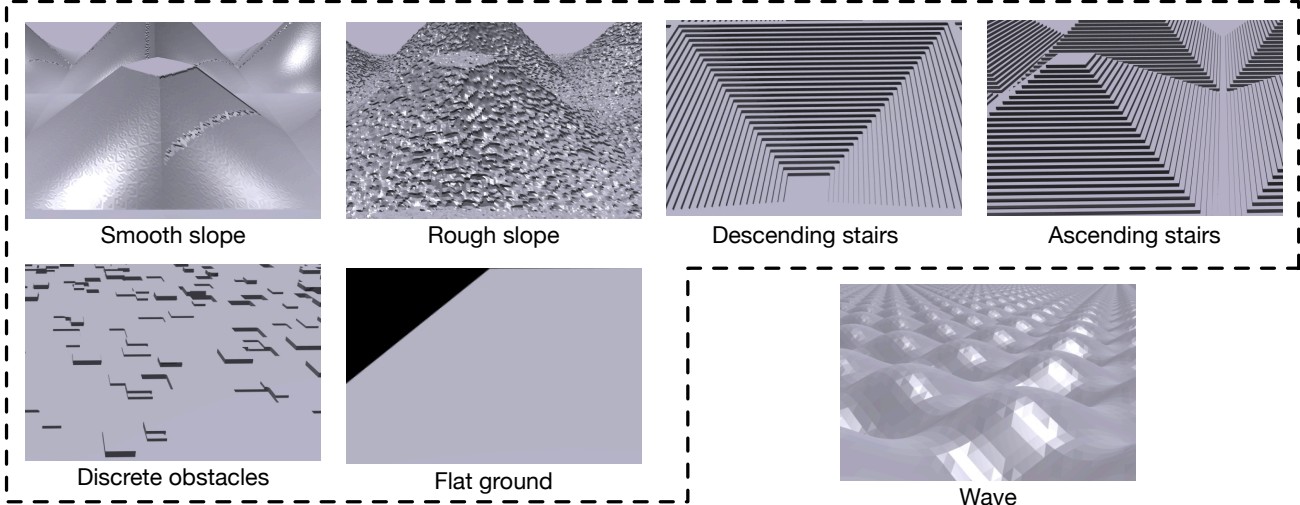

*Figure 8.* Visualization of different terrain types used in our experiments. Terrains enclosed by dashed boxes are used during training, while the wave terrain is reserved for out-of-distribution (OOD) evaluation.

## B.2. Terrains

To evaluate the robustness of interaction-conditioned policies under complex physical conditions, we augment all tasks with terrain-rich environments using the Isaac Gym framework, in contrast to prior work (Luo et al., 2024a; Peng et al., 2022) that primarily considers flat ground. Such environments induce diverse contact interactions and locomotion challenges. Specifically, the terrain types include *smooth slopes*, *rough slopes*, *staircases* (ascending and descending), *discrete obstacle ground*, and *flat ground*, sampled with proportions $\{0.2, 0.2, 0.2, 0.2, 0.1, 0.1\}$, respectively. By evaluating agents across a wide range of challenging terrains, this setup provides a stringent testbed for assessing the generalization and transferability of learned interaction policies across tasks. The visualization of different terrain geometries is shown in Figure 8.

*Table 8.* **Terrain types and sampling probabilities used for training and evaluation.** The environment includes six terrain categories with predefined sampling proportions to encourage robustness and generalization across diverse locomotion conditions.

| TYPES | PROBABILITY |
|---|---|
| SMOOTH SLOPES | 0.2 |
| ROUGH SLOPES | 0.2 |
| ASCENDING STAIRS | 0.2 |
| DESCENDING STAIRS | 0.2 |
| DISCRETE OBSTACLE GROUND | 0.1 |
| FLAT GROUND | 0.1 |

### B.3. Evaluation on Out-of-distribution Terrains

To evaluate generalization to structured but unseen terrain geometries, we construct a wave-shaped terrain with periodic height variations along both longitudinal and lateral directions. The terrain is generated procedurally by modifying the underlying height field using sinusoidal functions.

Concretely, given a base terrain grid, the height at each location is updated by superimposing a cosine wave along the forward direction and a sine wave along the lateral direction. The resulting height field can be expressed as:

$$h(x, y) = A \cos\left(\frac{y}{d}\right) + A \sin\left(\frac{x}{d}\right),  \tag{12}$$

where $A$ denotes the wave amplitude and $d$ controls the spatial frequency of the waves. The parameter $d$ is determined by the terrain length and the specified number of waves, ensuring a consistent number of oscillations across the terrain.

In our experiments, we set the number of waves to $10$ and the amplitude to $0.3$, scaled by the terrain vertical resolution. This produces smoothly varying surfaces without sharp discontinuities, preserving physical plausibility while introducing global geometric structures that are absent from the training distribution. An example visualization of the wave terrain is shown in Figure 8.

## C. Motion Reconstruction from Game Videos

We process the collected videos to obtain human motion representations. However, gameplay footage introduces challenges such as large viewpoint variations and partial occlusions. To mitigate these issues, we employ Grounding DINO (Liu et al., 2023; Ren et al., 2024) together with SAM2 (Ravi et al., 2024) to robustly segment and track the human subject from cluttered visual backgrounds. In practice, the reconstructed global root trajectories are often noisy, which can lead to physically implausible motions (Luo et al., 2024a). We therefore fix the root translation during subsequent learning. Finally, we utilize TRAM (Wang et al., 2024a) for 3D motion reconstruction from game videos, providing robust global trajectory and pose estimation under dynamic camera movements, commonly found in sports broadcasting. Specifically, TRAM estimates SMPL parameters (Loper et al., 2023) which include global root translation, orientation, body poses, and shape parameters and represent the estimated motions as proprioceptive states $s_t$ for downstream learning. To facilitate posterior inference in video domain, pseudo action labels are inferred from a pretrained motion imitation model(Luo et al., 2023).

## D. Experient on AI-Generated Videos

To demonstrate that TRIP is not limited to game footage, we construct an additional video dataset using two state-of-the-art text-to-video generation models (Seedance et al., 2026; Team et al., 2025). We generate 100 clips by prompting the models with descriptions of diverse terrain traversal scenarios, including slopes, stairs, uneven ground, and rocky surfaces. Each clip is approximately 10 seconds in duration.

Figure 9 shows representative frames sampled from the generated clips. Despite differences in visual style compared to game footage, the generated videos contain physically plausible human locomotion with clear motion variations, providing interaction diversity for primitive learning. This suggests a practical pathway toward scaling interaction primitive learning using AI-generated videos or real-world recordings of human in natural environments.

## E. MLLM-based Qualitative Evaluation

We conduct a case study to qualitatively assess the human-likeness of the learned behaviors using a multimodal large language model (MLLM) (Achiam et al., 2023). Specifically, we randomly sample two short clips from a single task and environment, generated by PULSE and our method (TRIP), respectively, without any method-specific annotations. The model is asked to perform a blind comparison between the two clips and determine which behavior appears more human-like, along with a brief justification based solely on observable motion characteristics.

As shown in Figure 10, the language model prefers the behavior generated by our method, citing smoother motion transitions, more consistent step-to-step adjustments, and better balance control over time. This case study provides qualitative evidence that our method may produce behaviors that are perceptually closer to human motion.

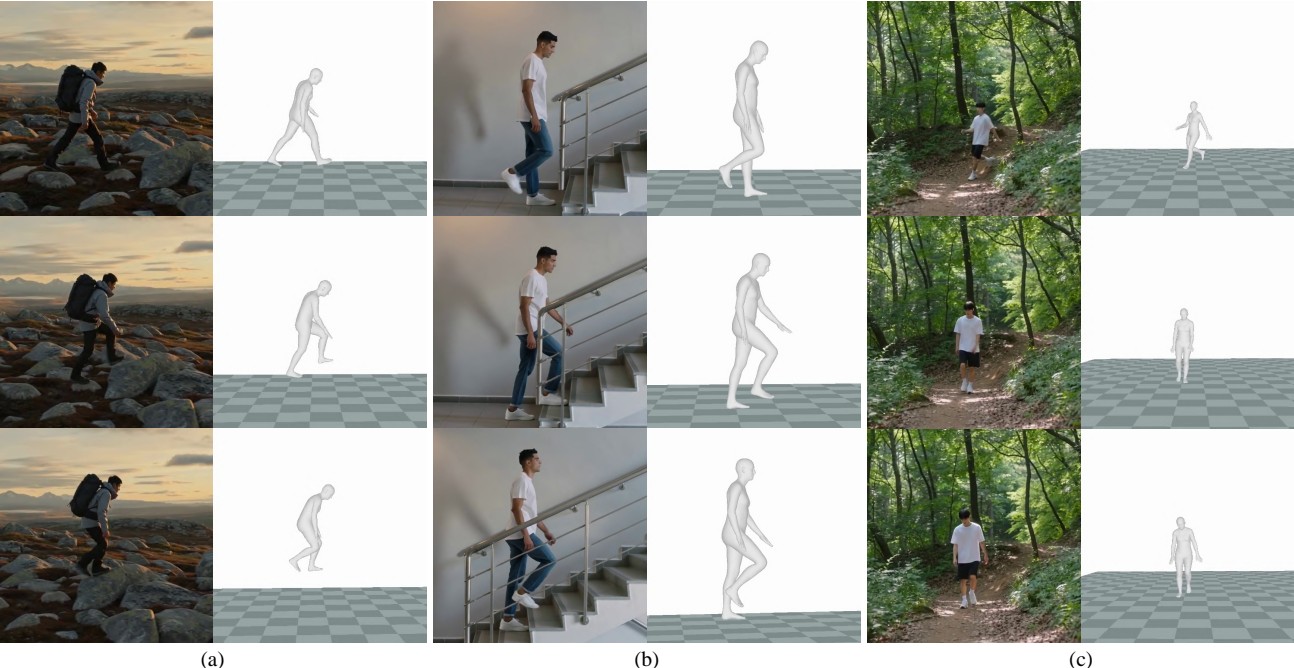

(a)                 (b)                 (c)

*Figure 9.* **Showcases of training video clips generated by text-to-video models.** (a) Walking over rugged mountainous terrain and rocky ground (by Seedance 2.0 (Seedance et al., 2026)) (b) Ascending staircases (by Kling 3.0 (Team et al., 2025)). (c) Moving downhill through jungle-like environments (by Seedance 2.0 (Seedance et al., 2026)).

## F. Implementation Details

We train TRIP with constant learning rate $2 \times 10^{-5}$. The discounted factor is $\gamma = 0.99$ and the clip range is $\epsilon = 0.2$. We leverage the proprioceptive prior, variational encoder and decoder from PULSE (Luo et al., 2024a) pretrained on AMASS dataset (Mahmood et al., 2019). The environment observation $h_t$ is the concatenation of the terrain heightmap and task inputs. The terrain observation is a rasterized local height map of size $\mathbb{R}^{32 \times 32 \times 3}$, which captures a $2 \times 2\ m^2$ square centered at the humanoid.

We conduct all experiments on a single RTX 4090 GPU with $1536$ parallel environments. We list the hyperparameters of TRIP in Table 9. The reinforcement learning hyperparameters are shared across all methods and adopted from PULSE (Luo et al., 2024a). The full optimization algorithm is in Alg. 1.

**(A) PULSE**                                    **(B) TRIP**

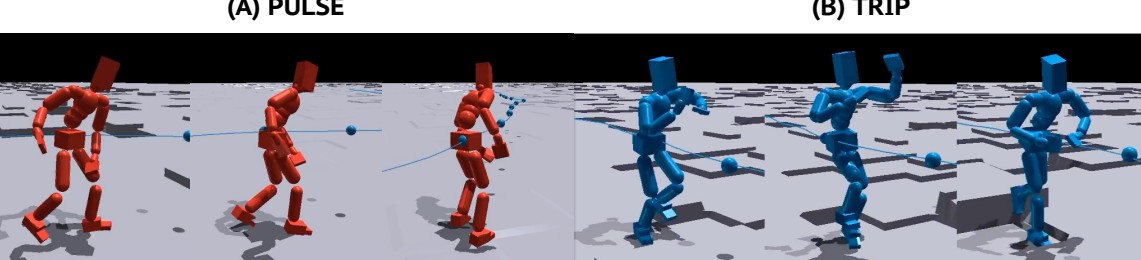

**Question:**

You are a human behavior evaluator for simulated humanoid actions. Your task is to compare two video clips (A and B) of a humanoid agent performing a task in a complex environment. These clips represent different methods of controlling the agent. You will evaluate the humanoid behaviors based on their **human-likeness** and provide a detailed justification for your choice.

**Please follow the steps below:**
1. Choose the more human-like behavior: Between Clip A and Clip B, which behavior seems more similar to a human?
● Choose either A or B. If neither, choose Tie.

2. Provide a brief justification: Based on the observable behavior in the clips, explain why you chose the more human-like option. Mention specific movement patterns, reactions, or recovery strategies you observed.

**Example Output:**
- **Choice**: A
- **Justification**: Clip A demonstrates more realistic walking patterns on uneven terrain, with smooth adjustments when encountering obstacles. The agent maintains better balance compared to Clip B, which exhibits unnatural foot sliding and instability.

Please follow these steps for each task and terrain. You will be given multiple pairs of clips to evaluate. After making all your decisions, you will be asked to provide aggregated results.

**Response:**

● Choice: B
● Justification: Clip B appears more human-like overall. The humanoid's movements are smoother and more consistent over time, with fewer abrupt changes between actions. Transitions between steps and body adjustments look more continuous, suggesting better balance control and anticipation−traits typical of human motion. In contrast, Clip A shows slightly more variability and abruptness in movement, which comes across as less natural, especially during ongoing motion and recovery between actions.

*Figure 10.* **Qualitative evaluation of human-likeness using MLLM.** The MLLM performs a blind comparison between two unlabeled short clips generated by PULSE and TRIP for the same task and environment, and selects the behavior that appears more human-like.

*Table 9.* Hyperparameters of TRIP.

| Name | Notation | Value |
|---|---|---|
| Interaction Imprimitive Codebook | | |
| Codebook size | $K$ | 256 |
| Codebook feature dimension | - | 128 |
| Commitment loss weight | $\beta$ | 0.25 |
| Sliding window horizon | $\tau$ | 2 |
| MLP $f_{enc}$ depth | - | 3 |
| MLP $f_{enc}$ width | - | 512 |
| MLP $f_{dec}$ depth | - | 3 |
| MLP $f_{dec}$ width | - | 512 |
| Activation | - | ReLU |
| Cross-Domain Physical Context Alignment | | |
| MLP $q_\phi$ depth | - | 3 |
| MLP $q_\phi$ width | - | $[512, 256, \text{out\_dim}]$ |
| MLP $p_\psi$ depth | - | 3 |
| MLP $p_\psi$ width | - | $[512, 256, \text{out\_dim}]$ |
| MLP $f_\xi$ depth | - | 3 |
| MLP $f_\xi$ width | - | $[512, 256, \text{out\_dim}]$ |
| MLP $f_r$ depth | - | 3 |
| MLP $f_r$ width | - | $[512, 256, \text{out\_dim}]$ |
| Context $c_t$ feature dimension | - | 128 |
| Activation | - | ReLU |
| Coefficient of $\mathcal{L}_{\text{alignment}}$ | $\lambda_{\text{alg}}$ | 0.1 |
| Coefficient of $\mathcal{L}_{\text{dynamics}}$ | $\lambda_s$ | 1 |
| Coefficient of $\mathcal{L}_{\text{reward}}$ | $\lambda_r$ | 1 |
| Cross-Domain Physical Context Alignment | | |
| MLP $\pi_\theta$ depth | - | 3 |
| MLP $\pi_\theta$ width | - | $[512, 256, \text{out\_dim}]$ |
| Activation | - | SiLU |
| KL coefficient of $\mathcal{D}_{\text{KL}}$ in $\mathcal{L}_{\text{reg}}$ | $\lambda$ | 0.1 |

