# OpenReview forum: "Learning Transferable Interaction Primitives from Game Videos for Humanoid Locomotion"
_ICML.cc/2026/Conference — ICML 2026 regular_

### Official Review · Reviewer_N7YQ · 2026-03-10

**Soundness:** 3
**Presentation:** 3
**Significance:** 3
**Originality:** 3
**Overall Recommendation:** 4
**Confidence:** 3

**Summary:**

This paper proposes TRIP (Transferable Interaction Primitives), a novel framework for learning humanoid control policies from unstructured, unlabeled game videos. Unlike prior methods that treat video as passive kinematic priors for motion imitation, TRIP explicitly focuses on extracting interaction primitives—discrete, short-horizon motor strategies that encode how humans dynamically interact with their environment. To bridge the domain gap between video and physical simulation, TRIP introduces a shared, domain-invariant latent space for "physical context," which is inferred from motion dynamics in videos and from explicit environmental observations in the target domain. This alignment enables the grounded execution of video-learned primitives in novel, complex terrains via reinforcement learning. Experiments on challenging locomotion tasks demonstrate that TRIP significantly outperforms baselines in terms of final performance, sample efficiency, and robustness to unseen terrains and external disturbances.

**Compliance With Llm Reviewing Policy:**

Affirmed.

**Ethical Review Concerns:**

1. Inappropriate Potential Applications & Impact (Human Safety Risk):

The method learns control policies from uncurated gameplay videos, which may contain unsafe, reckless, or self-destructive behaviors (e.g., jumping from great heights, crashing into obstacles). If such behaviors are encoded as “interaction primitives” and transferred to real-world humanoid robots without rigorous safety filtering, they could lead to physical damage, injury, or hazardous autonomous behavior. The paper mentions this risk only briefly in the Impact Statement but provides no concrete mitigation strategy (e.g., anomaly detection, safety constraints during execution, or behavioral validation).

2. Legal Compliance (Copyright and Terms of Use):

The use of commercial game videos (e.g., from titles like Mirror’s Edge) as training data raises potential copyright and End-User License Agreement (EULA) violations. Most game EULAs prohibit automated extraction of assets or motion data for commercial or research purposes without explicit permission. The authors do not disclose whether they obtained licenses, used royalty-free content, or conducted a legal review of their data sourcing pipeline—posing a risk of non-compliance with intellectual property law (e.g., DMCA, EU Copyright Directive).

3. Responsible Research Practice (Lack of Safety Documentation):

While the work is simulation-based, its stated goal is “real-world deployment.” Yet, there is no discussion of standard robotics safety protocols (e.g., ISO 10218, ISO/TS 15066), fail-safes, or human-in-the-loop validation that would be essential before physical deployment. This omission reflects insufficient attention to responsible research practices in embodied AI.

**Ethical Review Flag:**

Flag this paper for an ethics review.

**Key Questions For Authors:**

1. The term “interaction primitive” is central to your framework, yet it remains informally described. Could you provide a formal or operational definition?

2. Your method hinges on a shared latent context space $c_t$ bridging video-derived motion and simulation observations. What evidence do you have that this alignment is meaningful and not degenerate?

3. All experiments involve locomotion over static, known geometries. Does TRIP generalize to tasks requiring richer interactions—such as manipulating objects, navigating dynamic obstacles, or adapting to unknown surface properties (e.g., ice vs. mud)? If not, what are the fundamental limitations preventing such extension?

**Limitations:**

Yes. The authors adequately discuss the key limitations of their work, including its dependence on reliable video motion reconstruction and its current focus on terrain-centric locomotion.

**Strengths And Weaknesses:**

**Strengths**

1. The method is well-motivated and built upon a clear problem formulation that highlights the limitations of existing video-to-control approaches.

2. The proposed framework is modular, with distinct components for primitive learning, cross-domain context alignment, and policy learning, each supported by a specific objective function.

3. The experimental design is rigorous: it uses a diverse set of challenging terrains for training and evaluation, includes comprehensive comparisons against strong baselines (PPO, AMP, PULSE), and performs thorough ablation studies and generalization tests (e.g., scaled terrains, unseen wave-shaped geometry, external pushes).

4. The results consistently support the central claims of improved performance, efficiency, and robustness. The authors also honestly discuss the limitations, such as dependence on motion reconstruction quality.


**Weaknesses**

1. Core concept lacks formal definition and verifiability

The term “interaction primitive” is not formally defined as an observable or measurable physical/control variable. Instead, it is represented solely as discrete VQ-VAE codes, rendering it a black-box representation rather than a genuine model of environmental interaction. This makes it difficult to distinguish from high-level motion priors.

2. Cross-domain context alignment is unverified and theoretically questionable

The assumption that the video domain (inferred from reconstructed motion) and the simulation domain (ground-truth environment observations) share a common context latent space $c_t$​  is never quantitatively validated (e.g., via distributional distance or mutual information). Since videos cannot capture true physical properties (e.g., friction, material compliance), this alignment may be fundamentally ill-posed.

3. Experimental tasks are overly simplistic and lack genuine interactive scenarios

All experiments are confined to static, known-geometry terrains (slopes, steps, waves) and do not involve dynamic obstacles, deformable surfaces, object manipulation, or multi-agent settings—scenarios that truly require interactive strategies. Thus, the method’s applicability to “real-world deployment,” as claimed, remains unsubstantiated.

4. Baseline comparisons are insufficient, potentially overstating contributions

The paper omits comparisons with recent state-of-the-art video-to-control methods (e.g., VIMA, VIP, GamePlan) and strong sim-to-real baselines (e.g., Domain Randomization + RL). This ambiguity makes it unclear whether performance gains stem from the proposed “interaction modeling” or simply from using video data.

5. Two-stage training pipeline is decoupled and non-end-to-end

Primitive learning and policy training are performed separately, meaning the learned primitives may not be optimal for the downstream task, and the representation cannot be refined through task reward signals—limiting overall performance.

6. Sample efficiency evaluation is misleading

Efficiency is measured only in terms of RL environment steps, ignoring the substantial computational and data costs of video preprocessing (e.g., TRAM-based motion reconstruction and primitive library training). Total resource consumption is not reported, potentially inflating perceived efficiency gains.

7. Video data legality and safety risks are superficially addressed

While “unsafe behaviors” are mentioned, the paper does not describe how dangerous actions (e.g., jumping off cliffs) are detected or filtered. It also fails to discuss whether using commercial game footage violates copyright or end-user license agreements (EULAs), glossing over significant legal and ethical concerns.


---

Overall, this submission considers an important concept: moving beyond passive kinematic imitation to capture the active, dynamic interactions between an agent and its environment from video. This is a crucial step toward deploying robust humanoid controllers in the real world, where interaction with complex, unstructured environments is the norm. However, these weaknesses collectively indicate that while the paper presents an appealing vision, its current technical implementation and empirical validation are insufficient to fully support its core claims. Clarification or additional evidence would be necessary for confident acceptance at a top-tier venue.

---

> ### Author Rebuttal · Authors · 2026-03-31
>
> We appreciate the reviewer's valuable comments.
> > W1 & Q1. Interaction primitive.
>
> In our framework, interaction primitives are **latent motor strategies shaped by environment-dependent interaction dynamics**. Given motion trajectories, the primitive is constructed via: $s_t \xrightarrow{R} \tilde{z}\_t,\ s_{t:t+\tau+1} \xrightarrow{E} z_{t:t+\tau},\ g\_{k_t} = \arg\min_g \| f_{enc}([\tilde{z}\_t, z_{t:t+\tau}]) - g \|_2^2,\ (g\_{k_t}, \hat{z}_t) \xrightarrow{D} a_t$.
>
> It differs from conventional motion prior in two respects: (1) It encodes **short-horizon interaction dynamics**, not static poses. (2) Primitives are not passively sampled but **explicitly selected conditioned on physical context**.
>
> Although latent, they are not arbitrary: primitives correspond to decodable trajectories, and Figure 7 shows coherent usage across domains.
> > W2 & Q2. Domain alignment unverified.
>
> **Validity of context alignment.** The reviewer is correct that videos lack explicit force or material data. However, in TRIP, interaction context captures environmental constraints via **observable kinematics**. While physical parameters are unobservable, their aggregate effect on the kinematics $(s_t, a_t, s_{t+1})$ is observable. For instance, the specific trajectory of human balancing serves as a "sufficient statistic" for the underlying interaction logic. As formalized in the causal graphical model (Figure 2), $c_t$ is a domain-invariant latent mediator governing transitions.
>
> **Quantitative evidence.** We freeze $q_\phi$ and apply it to video data. A predictor using posterior $c_t$ achieves lower state and action prediction error (45.2%, 33.2%) than random $c_t$, demonstrating meaningful context extraction.
>
> Please refer to our response to Reviewer 4dir's W1 for details.
> > W3 & Q3. Tasks over simplistic.
>
> We acknowledge that the current work focuses on humanoid locomotion on diverse terrains. We will revise the manuscript to better scope our claims.
>
> While the tasks are terrain-focused, they go beyond simple static settings. OOD generalization and disturbance robustness (Tables 2–4) require adaptive interaction with varying and unseen dynamics, demonstrating non-trivial interaction capabilities.
>
> Additionally, following Reviewer 4dir's suggestion (W4 & Q3), we evaluated TRIP on **slopes with discrete obstacles**, where TRIP outperforms PULSE by 12.8–27.2%—demonstrating generalization to structurally distinct environments.
>
> More broadly, the TRIP framework is task-agnostic and extendable given appropriate video data with certain interaction. We will clarify these assumptions in the revision.
> > W4. Missing baselines.
>
> We thank the reviewer for suggesting these approaches. Our baselines follow the established standard in **humanoid control from motion/video data**. The suggested methods are non-comparable:
> - VIMA is a **multimodal** prompt-conditioned policy for tabletop manipulation.
> - VIP learns visual representations for **reward generation**.
> - Domain Randomization + RL addresses sim-to-real transfer, not video-to-control transfer.
>
> **Note:** We were unable to identify "GamePlan" in the video-to-control or humanoid locomotion literature. We would be grateful for a specific reference. We are open to including additional relevant baselines.
> > W5. Non-end-to-end pipeline.
>
> The two-stage pipeline is a reasonable design choice. The video domain provides a rich interaction structure, but no reward or environment state, so pretraining primitives yields a stable interface. Although primitives are fixed, the use is task-aware, since reward shapes primitive selection and residual adaptation.
> > W6. Sample efficiency ignores preprocessing.
>
> **(1) Standard definition in literature**
>
> In RL, sample efficiency is defined as the amount of **online environment interaction (samples)** required to reach a performance threshold, rather than wall-clock cost. Our evaluation follows this convention. This is because the primary bottleneck in deployment is **not GPU cycles** for preprocessing, but the physical time and safety risks associated with online samples. Under this metric, the comparison is fair, since all methods are trained under the same budget, and the reconstructed motions are used for both TRIP and the video-based baselines.
>
> **(2) A breakdown of offline and online costs**
>
> We also acknowledge that primitive learning adds offline cost; however, these are one-time representation-learning costs amortized across downstream tasks.
> || Training hours
> -|-
> Motion extraction|~3
> Primitive learning|~3.5
> Policy learning|~16
> > W7. Data legality and safety.
>
> The data usage follows guidelines by the game developer. Data is used for non-profit academic research; no raw content is redistributed. In addition, game data can be replaced by generated videos (see Reviewer nRSR's Q1).
>
> Regarding unsafe behaviors, the collected videos do not contain extreme actions. For broader application, unsafe segments can be filtered via manual or automated detection.

---

> > ### Author Rebuttal · Reviewer_N7YQ · 2026-04-07
> >
> > Thank you for the responses. I appreciate the commitments made, particularly around W1 (codebook analysis, visualization, mutual information with human labels) and W6 (total cost reporting). However, these are promises for the revision rather than results I can evaluate now. The core weakness — that "interaction primitive" remains a concept grounded primarily in downstream task performance rather than independently validated — is only partially addressed by the proposed analyses, which I have not yet seen. I will maintain my current assessment and defer final judgment to the revised manuscript.

---

> > > ### Author Response · Authors · 2026-04-07
> > >
> > > We appreciate the feedback and would like to clarify that the analyses mentioned in our previous response are not merely "promises". They have been completed, and we present the concrete results below to provide additional validation of the learned interaction primitives beyond downstream performance.
> > >
> > > ### **(1) Representation consistency**
> > >
> > > As shown in our response to Reviewer **4dir’s W1&Q1**, we analyze whether the learned discrete primitives exhibit consistent structure across domains. Using **Representational Similarity Analysis (RSA)**, we compute pairwise relationships between codes based on their induced state transitions and compare them across domains. Concretely, for each code, we compute its mean state-transition difference $\Delta s = s_{t+1} - s_t$ as a proxy for the interaction dynamics, and compare pairwise distance matrices across domains. We obtain a **Pearson correlation 0.789** and a **Spearman correlation 0.825**. These results suggest that the learned primitives capture consistent interaction structures. Notably, this is an intrinsic property of the representation beyond downstream RL performance.
> > >
> > >
> > > ### **(2) Training-time diagnostics**
> > >
> > > Furthermore, we monitor the primitive learning process using the following intermediate metrics:
> > > - Reconstruction error, $L_{\text{rec}}$ in Eq.(1), which reflects how well discrete codes preserve motion dynamics during training;
> > > - Codebook utilization (dead codes), which indicates whether the learned primitives are actively used or suffer from collapse;
> > > - Perplexity, $\exp(H(\text{usage distribution}))$, which measures the effective number of codes and thus the diversity of learned primitives.
> > >
> > > As shown in our response to **Reviewer nRSR's Q2** (ablation on codebook size $K$), these metrics provide clear evaluation for representation quality. In particular, increasing $K$ beyond 256 leads to a rise in dead codes and diminishing reconstruction gains, indicating over-capacity and under-utilization. Conversely, at $K=256$, we observe low reconstruction error, minimal dead codes, and high effective usage (~79.6%), suggesting a well-balanced and stable discrete representation.
> > >
> > > K|Recon Err(↓)|Dead codes|Perplexity
> > > -|-|-|-
> > > 64|0.397|4|36.4
> > > 128|0.308|5|87.2
> > > 256|0.265|4|203.7
> > > 512|0.255|52|369.9
> > >
> > >
> > > ### **(3) Functional validity via context grounding**
> > > While primitives are discrete, they are grounded through the latent context $c_t$, which captures the underlying interaction dynamics. We therefore evaluate whether this shared latent representation encodes predictive physical structure.
> > >
> > > As presented in our response to **Reviewer 4dir's W1&Q1**, we freeze the posterior $q_\phi$ and train a lightweight predictor to estimate $(s_{t+1}, a_t)$ from $(s_t, c_t)$. Compared to a control baseline using random $c_t$, the posterior-based representation reduces prediction error by 45.2% for state prediction and 33.2% for action prediction. These results indicate that the latent space underlying primitive selection encodes informative and predictive interaction dynamics.
> > >
> > > | |State MSE(x1e-3↓)|Action MSE(x1e-3↓)
> > > -|-|-
> > > Random $c_t$|8.18|0.869
> > > Posterior $c_t$|4.48|0.581
> > > Error reduction|**45.2%**|**33.2%**
> > >
> > >
> > > We provide these additional quantitative analyses to facilitate your evaluation, as manuscript revisions are not permitted during the rebuttal phase. We remain committed to incorporating these results into the final version.

---

### Official Review · Reviewer_1L68 · 2026-03-12

**Soundness:** 3
**Presentation:** 3
**Significance:** 3
**Originality:** 3
**Overall Recommendation:** 3
**Confidence:** 3

**Summary:**

The paper presents Transferable Interaction Primitives (TRIP), a framework for learning robust humanoid control from unstructured, unlabeled gameplay videos. Unlike traditional methods that focus on passive kinematic imitation, TRIP aims to capture dynamic humanoid-environment interactions by distilling a discrete library of interaction primitives from video motion dynamics. To bridge the gap between monocular video and physical simulation, the framework employs a shared, domain-invariant latent space that aligns physical context inferred from motion with context derived from environment observations. Experiments conducted in IsaacGym demonstrate that TRIP significantly enhances the robustness and generalization of humanoid policies across challenging terrains, such as slopes and stairs, compared to state-of-the-art reinforcement learning and motion-prior baselines.

**Compliance With Llm Reviewing Policy:**

Affirmed.

**Final Justification:**

I thank the authors for the rebuttal. However, given that some of my concerns remain unaddressed, I've decided to maintain my score.

**Key Questions For Authors:**

1. How would the library of learned primitives change if the framework were trained on a game with different physics or movement styles compared to the balance-heavy mechanics of the current source?

2. In the Trajectory task ablation studies, why did removing the interaction primitives lead to a much larger drop in success rate than disabling the cross-domain context alignment?

3. What is the impact of the codebook size on the model’s ability to generalize, and was there a specific reason for choosing 256 entries?

4. How does the receding-horizon execution strategy, which only uses the first latent variable, ensure temporal smoothness over longer, more complex interaction sequences?

5. Can the authors provide more detail on the value-difference-weighted regularizer and its specific role in balancing guidance from the video domain with the needs of reinforcement learning in the target domain?

**Limitations:**

yes

**Strengths And Weaknesses:**

Strengths

1. The approach demonstrates an innovative use of high-fidelity gameplay footage as a scalable data source, effectively bypassing the need for expensive and difficult-to-collect real-world robot interaction data.

2. The architectural decision to decouple "how" to move (via primitives) from "when" to move (via environmental context) provides a structured and logical path for transferring knowledge from the video domain to the physical control domain.

3. The resulting policies show impressive robustness and zero-shot generalization when tested on out-of-distribution terrain geometries and sudden external physical disturbances.

Weaknesses

1. There is a heavy dependence on the accuracy of third-party motion reconstruction and subject tracking tools; any noise or errors in the initial pose estimation can directly degrade the quality of the learned interaction primitives.

2. The current scope of the research is primarily limited to terrain navigation and locomotion, leaving the applicability of these interaction primitives to complex manipulation tasks largely unproven.

3. The training process relies on pseudo-action labels generated by a pretrained reference policy, which may limit the diversity and skill level of the learned agent to the performance ceiling of that specific teacher model.

---

> ### Author Rebuttal · Authors · 2026-03-31
>
> We appreciate the reviewer's valuable comments.
> > W1. Dependence on motion reconstruction.
>
> We acknowledge that any vision-based methods depend on input quality. However, TRIP is designed to improve robustness through a high-fidelity perception pipeline and a noise-resilient architecture.
>
> **Perception pipeline.** We use a robust ensemble of SOTA tools: TRAM for motion recovery, Grounding DINO for detection, and SAM2 for tracking, which are widely adopted tools providing reliable estimates.
>
> **Denoising via quantization.** TRIP does not directly replay reconstructed motions. It distills motion dynamics into a **discrete VQ codebook**, introducing an information bottleneck that reduces sensitivity to noise.
>
> **Refinement in the target domain.** The continuous residual $\delta_t$ (Eq. 7-8), learned via RL in the target domain, dynamically corrects minor kinematic inaccuracies from video-learned primitives.
> > W2. Limited to locomotion.
>
> Please refer to our response to Reviewer nRSR's W3. We acknowledge the locomotion focus and will refine the title in revision.
> >W3. Pseudo-action labels limit diversity to teacher ceiling.
>
> Pseudo-actions are used exclusively in the video domain to train the posterior inference $q_\phi$ and video policy $\pi_{video}$, **rather than as rigid supervision** for the final controller. The target policy $\pi_\theta$ can surpass the reference policy through:
> - $\pi_\theta$ is a standalone RL controller optimizing task rewards instead of **a direct clone** of the teacher model. It selectively invokes primitives based on the aligned context $c_t$.
> - The residual ($\delta_t$) enables further adaptation of context-conditioned primitives.
>
> If the reference policy were a hard ceiling, TRIP would at best match the motion-prior baselines. Instead, Table 2–4 show that TRIP consistently outperforms the baselines on complex terrains.
> > Q1. Different physics?
>
> Please refer to our response to Reviewer nRSR's W2 & Q1.
> > Q2. Primitives > alignment in ablation?
>
> We clarify that the interaction primitives and the context alignment serve distinct, hierarchical roles in our framework, and the ablation results in Figure 6 reflect this.
>
> **Why "w/o primitives" causes a collapse?** The interaction primitives form the **transferable content**—reusable motor strategies from video. Context alignment is built **on top of** the primitives. Without the underlying primitives, the alignment module has nothing meaningful to select from, forcing the policy to learn interactions from scratch.
>
> **Why "w/o alignment" causes a partial drop?** Without alignment, the policy still retains the structured primitive vocabulary, but loses the prior for associating primitives with the environment. In an RL setting, the agent can eventually "re-learn" some of these associations through trial and error, leading to degraded but not collapsed performance. The ablation in Figure 6 confirms that both components contribute.
>
> **The significance of the "minor" drop.** The performance gain from alignment is not "minor" in the context of transfer efficiency. While standard RL methods can eventually compensate for misalignment in an interactive training setting, the alignment is crucial for systematically grounding video-derived strategies. As shown in Figure 4, our method achieves substantially **faster adaptation and improved sample efficiency**.
> > Q3. Impact of codebook size.
>
> Please refer to our response to Reviewer nRSR's Q2.
> > Q4. Receding-horizon temporal smoothness.
>
> We argue that the receding-horizon execution in TRIP achieves a balance between **local kinematic coherence** and **global environmental reactivity**.
>
> First, although only the first latent variable $\hat{z}\_t$ is executed at each step, the decoder $f_{dec}$ is pretrained to reconstruct a short-horizon trajectory from a primitive $g_k$, so the latent $\hat{z}_t$ inherently encapsulates the subsequent intent.
>
> Moreover, during deployment, the humanoid moves to a new state at every step and receives updated terrain conditions. Executing the full decoded sequence open-loop would ignore these per-step changes.
>
> TRIP therefore executes only the first latent and replans at each step, allowing the policy to continuously correct its behavior based on refreshed proprioceptive feedback ($s_t$) and environmental context ($c_t$).
> > Q5. Value-difference-weighted regularizer.
>
> The regularizer acts as an adaptive distillation mechanism that accounts for both shared structure and cross-domain discrepancies. The value difference $\Delta_t$ serves as a relative advantage estimator:
> - When the video-derived policy is predicted to yield higher returns (e.g., encountering an interaction pattern seen in videos), we have $\Delta_t > 0$. The term encourages $\pi_\theta$ to align with $\pi_{video}$, transferring interaction knowledge.
> - When the RL policy is already estimated to perform better on target tasks, the constraint relaxes, reducing negative transfer from suboptimal behaviors.

---

> > ### Author Rebuttal · Reviewer_1L68 · 2026-04-04
> >
> > I appreciate the authors' detailed responses and the additional clarifications provided during the rebuttal phase. However, despite these technical clarifications, my primary concerns remain:
> >
> > 1. The framework's fundamental dependency on high-quality motion reconstruction continues to pose a risk to robustness in more diverse or occluded video settings.
> >
> > 2. While the authors have promised to refine the title and research scope, the current empirical evidence is still heavily limited to terrain-centric locomotion. The demonstrated generality of these primitives for broader, more complex humanoid interaction tasks (e.g., manipulation) remains unproven within the scope of this paper.

---

> > > ### Author Response · Authors · 2026-04-06
> > >
> > > We thank the reviewer for the continued engagement and address the remaining concerns.
> > > ### 1. Dependency on  motion reconstruction
> > > **(1) Motion reconstruction as a deliberate and favorable trade-off.**
> > > As the reviewer correctly noted, our approach uses *“gameplay footage as a scalable data source, effectively bypassing the need for expensive and difficult-to-collect real-world robot interaction data.”* This dependency on motion reconstruction is therefore intentional: it replaces reliance on costly robot interaction data with scalable video data. In this sense, it is not a limitation, but a **key enabler** that allows TRIP scalable without large-scale real-world data collection.
> > >
> > > Moreover, learning from video via motion reconstruction is a **common paradigm** in humanoid control[1-4], where human videos serve as demonstrations for control policies. Notably, PULSE learns motion priors directly from curated motion capture datasets, which impose even stronger data constraints. TRIP instead leverages readily available videos, providing a more flexible and scalable alternative. When motion capture data is available, our framework can also operate directly on such data, making motion reconstruction a replaceable interface.
> > >
> > > [1] ASAP: Aligning Simulation and Real-World Physics for Learning Agile Humanoid Whole-Body Skills. RSS 2025
> > >
> > > [2] SkillMimic: Learning Basketball Interaction Skills from Demonstrations. CVPR 2025
> > >
> > > [3] OKAMI: Teaching Humanoid Robots Manipulation Skills through Single Video Imitation. CoRL, 2024.
> > >
> > > [4] From generated human videos to physically plausible robot trajectories. arXiv, 2025.
> > >
> > > **(2) This dependency is not fragile in practice.**
> > > Beyond the architectural robustness discussed in W1 (VQ discretization and RL residual correction), we further emphasize three practical factors:
> > >
> > > - **Robustness to imperfect reconstruction.** To assess robustness to imperfect reconstruction, we inject noise into reconstructed poses. As shown in the following table, we observe only mild performance degradation, with TRIP consistently outperforming the baseline. This indicates that the method does not rely on precise kinematic accuracy. This is because with VQ discretization and RL residual correction, despite being noisy, these data still provide a meaningful starting point for policy learning.
> > >
> > > Noise scale|PULSE|TRIP
> > > -|-|-
> > > 0.02|-|235
> > > 0.05|-|228
> > > None|212|239
> > >
> > > - **Tolerance to occlusion.** Modern 3D pose estimation methods are reasonably robust to partial occlusion. We further validate this by two aspects: introducing synthetic occlusions (e.g., moving masks) as well as evaluating on naturally occluded videos (e.g., railings and foreground objects). We provide visualizations of motion reconstruction in the **link** (https://anonymous14159.github.io/trip), and we find that reconstructed motions remain physically plausible.
> > > - **Practical data conditions.** In our experiments, we use only ~20 minutes of video, making it practical to collect data with proper viewpoints and limited occlusion. Compared to trial-and-error RL in the target domain, this is a relatively lightweight requirement. In practice, severely occluded or corrupted clips can be easily filtered.
> > >
> > > **(3) Scalability across diverse video sources.**
> > > We further observe that TRIP performs well on AI-generated videos with diverse terrain and motion patterns (see response to nRSR, W2&Q1). The results are shown below, and we provide a visualization of the videos in the **link**. Despite variations in visual quality and occasional artifacts, the model still extracts consistent interaction primitives and achieves strong downstream performance. This suggests that the framework generalizes beyond curated gameplay footage and can leverage a broad range of video sources, further reducing reliance on robot interaction data.
> > > ||Trajectory|Speed|Reach
> > > -|-|-|-
> > > PULSE|212|238|147
> > > TRIP|232|243|170
> > >
> > > Overall, motion reconstruction introduces a dependency, but it is a favorable and scalable trade-off: it enables learning from abundant video data while remaining robust to realistic imperfections in practice.
> > >
> > > ### 2. Generality of primitives
> > > To further investigate whether the learned primitives generalize beyond terrain-based locomotion, we conduct an additional experiment on an object interaction task: box pushing. The goal is to move the box to the designated target position.
> > >
> > > We train our framework on AI generated box-pushing videos and adapt the context encoder to incorporate object-relative information, while keeping all other modules and the training pipeline unchanged. The results are shown in the **link**. TRIP achieves stable and more human-like behavior, outperforming the baseline.
> > >
> > > It provides initial evidence that the learned primitives are not limited to locomotion and have the potential to transfer to more interaction scenarios, which we identify as an important future direction.
> > >
> > > We sincerely hope these further clarifications and results can address your concerns.

---

### Official Review · Reviewer_nRSR · 2026-03-13

**Soundness:** 3
**Presentation:** 3
**Significance:** 2
**Originality:** 3
**Overall Recommendation:** 4
**Confidence:** 4

**Summary:**

TRIP learns humanoid locomotion policies from Death Stranding gameplay footage. The method first reconstructs human motion from 116 video clips using TRAM, then quantizes these motions into 256 discrete interaction primitives via VQ-VAE. Each primitive encodes a short-horizon movement pattern (e.g., a downhill gait) but carries no information about the environment that produced it. To deploy these primitives in a physics simulator, TRIP learns a shared latent "physical context" variable: in the target domain (IsaacGym), a context encoder reads a privileged terrain heightmap; in the video domain, a posterior model infers the same context from state transitions alone. KL divergence aligns the two, with auxiliary transition and reward prediction losses preventing collapse. A hierarchical RL policy then selects which primitive to execute based on the inferred context and adds a continuous residual for fine-grained adjustment. On three locomotion tasks across six terrain types, TRIP outperforms PPO, AMP, PULSE, and PULSE+AMP, with particular gains on harder terrains, unseen terrain shapes, and under external disturbances.

**Compliance With Llm Reviewing Policy:**

Affirmed.

**Final Justification:**

This work separates interaction primitives from physical context, which I think is a well-motivated design for video-to-control transfer. The probabilistic framework is clean and the ablation isolates each component well.
The rebuttal covered my main concerns.

Limitations include no real-world footage, sim-to-real gap unvalidated, and locomotion only, but the authors acknowledge them and commit to narrowing the claims in revision.

I keep my score at 4 (Weak Accept). The idea is strong, the rebuttal was responsive.

**Key Questions For Authors:**

1. *Why game videos?* The paper uses only Death Stranding but doesn't explain why game videos are needed. I'd like the authors to address: (a) Which property matters most? The dense terrain coverage, the clean character rendering, or the game engine's physically-plausible motion? (b) The posterior model assumes state transitions reflect terrain physics. In real videos, gait is confounded by fatigue, footwear, habits, social context. How robust is context inference to these? (c) A small experiment with a second source, e.g., different game, or even real hiking video, would go a long way. If the method only works on game footage, that should be stated as a scope constraint rather than left implicit.
2. K=256 with 116 clips (~35k frames) means ~137 frames per code on average. How many codes are actually used? Is there a long tail of dead codes? An ablation over K (64, 128, 256, 512) on one task would clarify whether 256 is justified or just a default.
3. The context encoder needs privileged heightmaps. What would it take to replace this with noisy depth camera input or proprioceptive history? Have you tried adding noise to the heightmap to simulate sensor imperfections?

**Limitations:**

Yes, section 7 acknowledges the locomotion focus and dependence on motion reconstruction quality.

**Strengths And Weaknesses:**

**Strengths:**
- The separation of "what motion to perform" (primitives) from "when to perform it" (context) is the right abstraction for this problem. Prior methods like AMP and PULSE treat video as a flat kinematic prior; they encode poses but ignore the fact that terrain shapes the motion. TRIP's insight into inferring environment context from dynamics alone (without seeing the terrain) and aligning it with privileged observations in simulation is well-motivated. The value-difference-weighted bidirectional KL (Eq. 11) for selective distillation from the video policy is also a careful design choice.
- Extracting interaction knowledge from passive video is a problem worth solving. The fact that 19 minutes of Death Stranding footage yields primitives that help a simulated humanoid handle terrains it was never explicitly trained on is a promising signal. The improvements over PULSE (which has the same AMASS-derived motion prior but no context conditioning) isolate the value of modeling interactions.

**Weaknesses:**
1. The generalization tables (Tables 2-4) report single numbers with no variance. Some gaps are modest, e.g., TRIP 196 vs. PULSE 171 in Table 2 Trajectory, a 14.6% margin. Without knowing the seed-to-seed variance, it's hard to tell if this is statistically reliable.
2. Everything comes from 116 clips of one game. Death Stranding is terrain-heavy by design, as the entire game is about traversing difficult landscapes. The abstract says "unstructured, unlabeled game videos," but these are neither unstructured nor representative. Would the pipeline work on a different game with less terrain variety?
3. The title says "interaction primitives" but the experiments only test locomotion. The framing promises more than the experiments deliver.
4. Line 017 frames the work around "real-world deployment," but there is no sim-to-real discussion anywhere. The context encoder requires privileged 32×32×3 terrain heightmaps (Appendix E). On a real robot, you'd need depth sensing plus accurate localization to produce anything comparable. Section 7 acknowledges the locomotion scope but says nothing about this deployment gap.

---

> ### Author Rebuttal · Authors · 2026-03-31
>
> We appreciate the reviewer's valuable comments.
> > W1. No variance in Tables 2-4.
>
> The results in Tables 2–4 are averaged over multiple episodes. We further evaluate seed-to-seed variance. As shown in the following tables, we found that the standard deviation is consistently within 5% of the mean. We will include variance in the revision.
>
> Increased difficulty.
> Model|Trajectory|Speed|Reach
> -|-|-|-
> AMP|153.6±2.9|157.5±2.9|5.9±0.6
> PULSE|172.6±14.2|196.5±6.8|97.2±3.7
> PULSE+AMP|161.9±14.2|180.2±2.8|5.2±1.5
> TRIP|**197.3±3.4**|**216.1±3.4**|**123.8±8.2**
>
> Unseen terrain.
> Model|Trajectory|Speed|Reach
> -|-|-|-
> AMP|160.8±9.8|134.1±22.1|6.1±1.9
> PULSE|180.2±7.6|214.2±8.0|114.7±2.3
> PULSE+AMP|173.2±6.8|210.8±10.0|5.4±0.8
> TRIP|**223.4±11.9**|**232.9±7.2**|**183.3±1.8**
> > W2 & Q1. Single game source. Which property matters most? Experiment with a second source?
>
> We thank the reviewer for the insightful questions on the choice of game videos and the generality of our approach.
>
> **(a) "Unstructured" does not imply arbitrary or random footage.**
>
> First, we clarify that "unstructured" denotes the absence of interaction, terrain, or contact annotations--all interaction primitives are learned in an unsupervised manner.
>
> **(b) Why game videos?**
>
> Game videos serve as a controlled yet diverse testbed, where the key factor is **interaction diversity induced by terrain variation**, rather than the game itself. The properties raised by the reviewer all contribute to the framework. We find terrain-induced interaction diversity and dynamically consistent motion to be the most critical, while clean rendering matters only for motion reconstruction. Modern physics-driven game engines naturally provide these properties, making them a strong and scalable testbed for studying interaction transfer. Our framework itself is source-agnostic and does not depend on any specific game.
>
> **(c) Generality across data sources.**
>
> Following the reviewer's suggestion, we additionally train on **AI-generated videos** (100 clips with diverse terrain and motion patterns generated by Seedance 2.0 and Kling). We observe consistent downstream performance gains, indicating independence from a specific game engine or dataset.
> ||Trajectory|Speed|Reach
> -|-|-|-
> PULSE|212|238|147
> TRIP|232|243|170
> > W3. Title says "interaction primitives" but only tests locomotion.
>
> We acknowledge that the current work focuses on **locomotion**. In this context, "interaction primitives" refer to motor strategies shaped by the environment's terrain geometry, i.e., how the humanoid adapts its gait and balance in response to different environment geometry. To avoid ambiguity, we will tighten the manuscript to clearly reflect our focus.
>
> Meanwhile, we believe that the TRIP framework (including primitive learning, cross-domain context alignment, and context-guided policy) is task-agnostic. With appropriate video data, it can be extended to broader humanoid tasks.
> > W4 & Q3. No sim-to-real. Privileged heightmaps. Robustness to noisy input?
>
> We understand the reviewer's concern for the deployment gap. Our use of privileged heightmaps follows prior work such as PULSE, which is a standard input process.
>
> To assess robustness to imperfect perception, we inject Gaussian noise into the heightmap during inference on the Trajectory task. The results are shown below:
> Noise scale|PULSE|TRIP
> -|-|-
> 0.01|207|235
> 0.05|211|230
> None|212|239
>
> TRIP consistently outperforms PULSE under all noise levels, with only mild degradation, indicating robustness to perception noise.
>
> Our contribution is the interaction transfer framework, which is compatible with other perception modalities. The heightmap can be replaced by depth sensing without changing the core method (e.g., ego-centric depth map in [1]). We will revise the wording to clarify the contribution and acknowledge bridging sim-to-real in detail as an important direction for future work.
>
> [1] Hiking in the Wild: A Scalable Perceptive Parkour Framework for Humanoids.
> > Q2. Codebook utilization and K ablation.
>
> We appreciate the reviewer’s suggestion and ablate the hyperparameter $K \in$ {$64, 128, 256, 512$} (codebook size) during the primitive learning. We report:
> - Reconstruction error ($L_{\text{rec}}$ in Eq. 1), which measures how well discrete codes preserve the original motion dynamics;
> - \#Dead codes, which is the number of codes not activated;
> - Perplexity $\exp(H(\text{usage distribution}))$, which reflects the effective codebook size.
>
> K|Recon Err(↓)|Dead codes|Perplexity
> -|-|-|-
> 64|0.397|4|36.4
> 128|0.308|5|87.2
> 256|0.265|4|203.7
> 512|0.255|52|369.9
>
> The results show that $K=256$ is the sweet spot: reconstruction error drops substantially from $K=64$ to $K=256$ but nearly saturates beyond this point. Meanwhile, dead codes remain minimal at $K=256$, whereas $K=512$ produces 52 dead codes, indicating the data does not support that level of granularity. Perplexity results confirm the codebook is well-utilized, with an effective usage rate of 79.6%.

---

> > ### Author Rebuttal · Reviewer_nRSR · 2026-04-01
> >
> > I appreciate the authors' effort in this rebuttal. A few follow-up points:
> > - The AI-generated video experiment is a nice addition. Consider sharing these videos via an anonymous link so reviewers can judge their quality. Real-world footage is still untested, but understandable given the rebuttal timeline. In revision, please make explicit that "unstructured" refers to the lack of interaction annotations, not arbitrary footage.
> > - The noise injection result is useful, though adding Gaussian noise to a ground-truth heightmap is quite different from real depth-camera input (occlusion, registration errors, partial observability). I'd suggest discussing the sim-to-real gap more concretely and positioning real-world deployment as future work.
> > - The authors claim the framework is "task-agnostic" and extendable to broader humanoid tasks. Could you be more specific? Would TRIP handle tasks involving hand-object contact, like pushing, carrying, and door opening? What video sources would provide the interaction primitives for those, and would the context encoder need changes? A brief discussion of what "broader tasks" concretely means would make the task-agnosticism claim easier to evaluate.
> >
> > ---
> > (update)
> > Thanks for the authors' additional clarifications. My concerns are resolved. I keep my positive score.

---

> > > ### Author Response · Authors · 2026-04-03
> > >
> > > Thank you for your timely follow-up and constructive suggestions.
> > >
> > > **1. AI-generated videos & “unstructured” terminology.**
> > >
> > > We have provided an anonymous **link** to the AI-generated videos for reference:
> > > https://anonymous14159.github.io/trip.
> > >
> > > We agree that the term “unstructured” may be ambiguous. In the revision, we will explicitly clarify that it refers to the absence of interaction annotations in the video footage, not that the footage itself is arbitrary or unconstrained.
> > >
> > > **2. Sim-to-real considerations.**
> > >
> > > We agree with the reviewer that injecting Gaussian noise into a ground-truth heightmap only partially simulates real-world perception challenges. As the reviewer points out, depth-based sensing introduces additional issues beyond additive noise, including:
> > >
> > > - Occlusion and missing depth values (e.g., on reflective or transparent surfaces),
> > >
> > > - Registration errors between depth maps and the assumed ground plane,
> > >
> > > - Partial observability due to limited sensor field of view or line‑of‑sight constraints,
> > >
> > > - Calibration inaccuracies and sensor‑specific noise patterns.
> > >
> > > We acknowledge that our Gaussian noise injection is a simplified, preliminary robustness check rather than a faithful simulation of real depth input. In the revision, we will explicitly discuss these sim‑to‑real gaps and clarify that a systematic treatment of the issues, along with real‑world deployment, is an important direction for future work.
> > >
> > >
> > > **3. On task-agnosticism.**
> > >
> > > We appreciate the chance to clarify this point. Extending TRIP to tasks involving hand-object interactions (e.g., pushing, carrying, or door opening) would require two main changes:
> > >
> > > - Video sources: These would need to contain rich hand-object interactions relevant to the target tasks (e.g., manipulation datasets or gameplay videos involving object use). This ensures that the discovered interaction primitives correspond to meaningful behaviors, such as applying force to an object, grasping, or articulating a hinge.
> > >
> > > - Context encoder adaptation: The encoder would need to shift from terrain-centric inputs (e.g., heightmaps) to object-centric representations, capturing geometry and interaction-relevant features. For instance, pushing/carrying would require capturing object geometry and contact location relative to the robot, while door opening would involve capturing kinematic constraints (articulated motion, handle position), or just using visual input as a proxy. In all cases, the context encoder's role is to produce an estimate of the current task-relevant state from raw perception, which is then used to select and ground discrete interaction primitives.
> > >
> > > Importantly, these changes affect only the input representation of the context encoder. The rest of the modules and the core framework — learning discrete interaction primitives and grounding them across domains — remain unchanged. However, the observation space (what the encoder consumes) must be chosen appropriately for the target task category.
> > >
> > > Again, thank you for encouraging us to clarify these points.
> > >
> > > ---
> > > **[Update Apr. 7th]: Additional experiment on object interaction.**
> > >
> > > We appreciate the reviewer's positive assessment and are glad that the concerns have been resolved in the previous discussion.
> > >
> > > As a further step to support the method's generality — and to more directly address the reviewer's earlier question — we have implemented the design discussed above on an object interaction task: box pushing. The goal is to move the box to a designated target position.
> > >
> > > Concretely, we use AI-generated box-pushing videos as source data, and adapt the context encoder to incorporate object-relative information (e.g., box position). The results are shown in the **link** (https://anonymous14159.github.io/trip). TRIP achieves more stable and human-like behaviors, outperforming the baseline.
> > >
> > > While this experiment is conducted in simulation rather than with real perceptual inputs, it provides initial evidence that the learned primitives are not limited to locomotion and have the potential to transfer to broader interaction scenarios. Bridging the gap to real perceptual inputs and more complex tasks is an open direction worthy of future investigation.

---

### Official Review · Reviewer_4dir · 2026-03-19

**Soundness:** 3
**Presentation:** 2
**Significance:** 3
**Originality:** 3
**Overall Recommendation:** 5
**Confidence:** 3

**Summary:**

This paper introduces Transferable Interaction Primitives (TRIP), a framework focused on learning robust humanoid control policies from passive, unstructured game videos to navigate complex physical environments. Existing video-imitation methods typically extract passive kinematic motions without understanding the underlying environmental context. To make these video-mined primitives environment-aware, TRIP align a shared latent space, which is achieved by matching the physical context implicitly inferred from video motions with the explicit environmental observations in the target robotic domain. Experimental evaluations on locomotion tasks show TRIP's enhanced robustness to both unseen terrain geometries and unexpected external disturbances.

**Compliance With Llm Reviewing Policy:**

Affirmed.

**Final Justification:**

The TRIP framework proposed in this paper goes beyond traditional passive motion imitation by learning transferable motion primitives, demonstrating strong originality and significance. During the rebuttal phase, the authors provided good responses to the key questions I raised in the initial review. Most notably, regarding my primary concern about the generalization of $q_\phi$, the authors supplemented their response with a compelling predictor experiment, demonstrating its effectiveness in extracting physical semantics within the video domain. Furthermore, the additional Representational Similarity Analysis and the evaluation on complex OOD terrains further solidified the paper's conclusions. In summary, I am raising my final evaluation from Weak Accept to Accept.

**Key Questions For Authors:**

1. Could the authors clarify how q_phi reliably generalizes to video dynamics without seeing video data during the context alignment phase? Are there underlying assumptions about the shared manifold of simulation and video kinematics that guarantee this zero-shot transfer? This is my primary concern, and it would validate the core "cross-domain alignment" claim and significantly raise my score.

2. Given the codebook size of 256, is there quantitative evidence (e.g., cross-domain retrieval accuracy, mutual information, or confusion matrices) demonstrating that the same code indices consistently map to the identical physical interactions across the two domains on a global scale?

 3. Have the authors evaluated TRIP on structurally distinct OOD geometries that cannot be approximated by training set interpolations (e.g., discontinuous gaps, slope with obstacles)?

**Limitations:**

Yes

**Strengths And Weaknesses:**

Strengths:

1. Novel Interaction-Centric Paradigm: The paper cleverly shifts the focus from traditional passive motion imitation to explicitly modeling humanoid-environment interactions.

2. Strong Empirical Performance: The TRIP framework achieves impressive results across multiple challenging, terrain-rich tasks. It consistently outperforms strong baselines like PULSE and AMP in terms of sample efficiency.

Weaknesses:

1. Unjustified Zero-Shot Context Alignment: In Algorithm 1, the posterior q is trained exclusively on target-domain simulation data (Step 2), but is then used zero-shot to infer physical context from game videos (Step 3). The assumption that q can perfectly generalize to out-of-domain video dynamics is unconvincing and lacks justification.

2. Insufficient Proof of Alignment: The qualitative visualization in Figure 7 only demonstrates alignment for 2 semantic categories. Given the codebook size of 256, this is too coarse to rigorously validate the overall cross-domain alignment effect.

3. Misattributed Performance Gains: The ablation study (Figure 6) reveals that removing the alignment module causes only a minor performance drop, and the model still significantly outperforms the PULSE baseline. The severe drop in the "w/o primitives" setting indicates that the performance gain predominantly stems from the video-pretrained primitives, not the proposed alignment mechanism.

4. Weak Out-of-Distribution (OOD) Evaluation: The "wave-shaped terrains" in Table 3 are claimed to be unseen geometries. However, since they are generated using sinusoidal functions , they essentially act as spatial interpolations of the "smooth slopes" already heavily sampled during training, weakening the OOD generalization claim.

5. Risk of Reward Hacking: The Reach task reward relies solely on the Euclidean distance of the right hand to the target, lacking stability constraints. The authors fail to empirically demonstrate that the learned primitives alone can prevent reward-hacking behaviors (e.g., adopting unnatural postures to minimize hand distance).

---

> ### Author Rebuttal · Authors · 2026-03-31
>
> We appreciate the reviewer's valuable comments.
> > W1 & Q1. How does $q_\phi$ generalize? What justifies the transfer?
>
> We clarify that the transferability of $q_\phi$ is grounded in the causal invariance of the interaction context $c_t$ across domains.
>
> **Design rationale:** As shown in the causal graph (Figure 2), the physical context $c_t$ acts as a latent mediator that governs state transitions across both domains. The posterior $q_\phi$ infers $c_t$ from kinematic transitions $(s_t, a_t, s_{t+1})$, consistently well-defined across domains via the shared SMPL representation. Specifically:
> - Universal interface: Using SMPL representation as a shared kinematic interface, $q_\phi$ operates on normalized states $s_{t:t+1}$. This significantly reduces the domain gap compared to raw visual inputs, as both domains share the same underlying articulated skeleton and kinematic constraints.
> - Objective: Crucially, the **alignment loss (Eq. 6)** explicitly regularizes $c_t$ to encode physically predictive factors that are consistent across domains, ensuring that the latent space is anchored to fundamental motion dynamics.
>
> **Empirical verification:** To validate whether $q_\phi$ produces meaningful context in the video domain, we freeze it and apply it to video-domain motion data. A lightweight predictor is trained to predict the next state $s_{t+1}$ and action $a_t$ from $[s_t, c_t]$. As a control, we replace $c_t$ with random vectors.
> | |State MSE(x1e-3↓)|Action MSE(x1e-3↓)
> -|-|-
> Random $c_t$|8.18|0.869
> Posterior $c_t$|4.48|0.581
> Promotion|**45.2%**|**33.2%**
>
> The significant reduction indicates that $c_t$ encodes structured physical semantics from video-domain transitions. Since the predictor has no access to the actions or future states, this gain stems from the high-quality physical context inferred by $q_\phi$.
> > W2 & Q2. Quantitative alignment evidence beyond Figure 7?
>
> We provide architectural and quantitative evidence:
>
> **Architectural consistency** The codebook $\mathcal{G} = \{g_k\}$ and decoder $f_{\text{dec}}$ are **shared and frozen** across domains, so each code always decodes to the same moton primitive regardless of its source.
>
> **Quantitative consistency.** Without paired ground-truth labels, we use **Representational Similarity Analysis (RSA)** to assess cross-domain alignment. For each code, we compute its mean state-transition difference $\Delta s = s_{t+1} - s_t$ as a proxy for the interaction dynamics, and compare pairwise distance matrices across domains. We obtain Pearson correlation 0.789 and Spearman correlation 0.825, indicating that the relative geometric structure among codes is well-preserved. This provides quantitative evidence that the codebook consistently organizes interaction semantics across domains.
> > W3. Misattributed performance gains.
>
> Please see responses to Reviewer 1L68's Q2.
> > W4 & Q3. Weak OOD evaluation. Any evaluation on structurally distinct OOD geometries?
>
> First, we clarify that wave-shaped terrains are not spatial interpolations of smooth slopes; rather, they present a fundamentally different dynamic control regime.
> - Slope terrains are monotonic with a consistent gradient, requiring **sustained force and quasi-static progression**.
> - In contrast, wave terrains form **low-amplitude, uneven, undulating surfaces** with rapid local variations, which require **continuous re-balancing and adaptive foot placement**.
>
> Empirically, a policy trained on slopes fails to maintain balance on wave terrains, confirming that the latter represents a distinct dynamic disturbance rather than a simple interpolation.
> Terrain|Height range|Pattern|Gradient
> -|-|-|-
> Wave|-0.6~0.6m|periodic, wavelength 0.2 m|varying
> Slope|-9.7~9.7m|monotonic rise to center|consistent
>
> Following the reviewer’s suggestion, we evaluate TRIP on a more challenging and discontinuous environment: **slopes with discrete obstacles**. These obstacles introduce abrupt height changes that cannot be approximated by smooth interpolation. The results are:
> Method|Trajectory|Speed|Reach
> -|-|-|-
> PULSE|148|145|80
> TRIP|**167(+12.8%)**|**171(+17.9%)**|**112(+27.2%)**
>
> TRIP significantly outperforms the PULSE baseline across all metrics. This gain is attributed to TRIP’s ability to recognize the "obstacle" context via alignment and invoke appropriate interaction primitives, demonstrating generalization to structurally distinct OOD environments.
> > W5. Reward hacking in Reach.
>
> The Reach reward is intentionally task-oriented, and we follow this standard protocol used in prior work (e.g., AMP, PULSE), enabling fair comparison. That said, TRIP naturally mitigates the risk of reward-hacking through its structure. TRIP operates through interaction primitives derived from video and a pretrained motion decoder, which acts as a strong kinematic prior, constraining the policy to produce human-like motions. Figure 5 shows natural reaching postures during execution, and TRIP outperforms baseline in the human-likeness study in Figure 10.

---

> > ### Author Rebuttal · Reviewer_4dir · 2026-04-03
> >
> > Thanks for your rebuttal. I would raise my evaluation to Accept.

---

> > > ### Author Response · Authors · 2026-04-04
> > >
> > > Thank you again for your insightful comments and suggestions. We are very glad that our responses have addressed your concerns, and we sincerely appreciate your positive feedback and your willingness to raise the score.
> > >
> > > In the final version, we will incorporate the requested clarifications and reflect your suggestions in both the main paper and the appendix.
> > >
> > > *Just a small note: it seems that the updated score may not yet be reflected on our side—this might be a display delay, but we mention it here in case it is helpful.*
> > >
> > > Thank you once again for your time and effort in helping us improve the manuscript!

---

### Review · Ethics_Reviewer_fJ2U · 2026-03-20

**Recommendation:** Remediation action needed

**Ethics Issue:**

This paper raises three ethical concerns under the conference's ethics guidelines.

1. Legal Compliance — Copyright, Intellectual Property, and Data Licensing
The paper uses video footage from a video game produced by Kojima Productions as source data. The authors do not indicate whether they obtained consent from Kojima Productions to use this material for research purposes, do not discuss the licensing conditions or terms of use applicable to the footage, and do not document the data source in sufficient detail for readers to assess its provenance or legal status. The conference guidelines require that "data sources, licences, and terms of use are clearly stated and respected" and that "submissions utilising datasets that contain copyrighted material should acknowledge and address this in the impact statement." Neither requirement is met.

2. Inappropriate Potential Applications & Impact
The paper presents a framework for deriving interaction primitives from video game motion sequences to control humanoid robots. The authors' impact statement acknowledges beneficial applications such as search and rescue and assistive robotics, but makes no mention of potential harmful applications. A framework capable of generating human-like motion primitives for humanoid robot control has foreseeable harmful applications, including deployment in autonomous weapons systems, physically coercive operations, or other high-risk contexts. The conference guidelines require that "papers about applications that have a direct connection to human rights issues should provide a thoughtful discussion of the risks of the application." The current impact statement is one-sided and incomplete in this respect. Please note that potential military or defence applications raise independent concerns under international humanitarian law and ongoing international discussions on Lethal Autonomous Weapons Systems (LAWS) under the Convention on Certain Conventional Weapons (CCW).

3. Bias, Safety, and Responsible Research Practice
The authors themselves acknowledge that use of uncontrolled videos may cause the system to inherit biases or unsafe behaviours from the source and note that careful data curation, safety constraints and similar measures are needed for "real-world settings." However, having identified this risk, the authors provide no explanation of what steps, if any, they took to address it in their own work. There is no discussion of the criteria used to select video game sequences for inclusion as interaction primitives, no explanation of how unsafe or biased behaviours were identified and excluded from the dataset, and no reference to standard robotics safety protocols and their relevance. Given that this framework is explicitly intended for deployment in humanoid robotics — a context with direct human safety implications — this is a significant omission that undermines confidence in the safety of the proposed approach.

**Remediation Action:**

On copyright and data licensing:
- The authors should disclose whether consent was obtained from Kojima Productions or other relevant private studios to use their game footage for this purpose, and under what terms.
- The licensing conditions and terms of use applicable to the footage should be clearly stated, with confirmation of compliance.
- The data source should be properly cited and documented, including the specific title, scope of footage used, and method of collection.

On misuse and broader impact:
- The impact statement should be revised to include a balanced and honest discussion of potential harmful applications of this framework, including but not limited to autonomous weapons systems and other physically harmful deployments.
- For potential military or defence applications, the authors should acknowledge the relevant international regulatory context, including LAWS discussions under the CCW.
- Where feasible, the authors should propose concrete mitigation strategies or scope limitations to reduce the risk of harmful applications.

On bias, safety, and data curation:
- The authors should explain what criteria were used to select video game sequences for inclusion, and how sequences likely to introduce unsafe or biased behaviours were identified and excluded. If no such process was undertaken, this should be explicitly acknowledged as a limitation.
- Given the explicit acknowledgement of safety risks in real-world deployment, the authors should reference and engage with standard robotics safety protocols relevant to their framework.
- If systematic safety evaluation was not performed as part of this work, the authors should clearly state this, discuss the implications for real-world deployment, and recommend what steps future work or practitioners should take before deploying systems derived from this framework.

---

### Decision · Program_Chairs · 2026-04-30

**Decision:**

Accept (regular)

**Comment:**

This paper proposes TRIP, a framework for learning transferable humanoid interaction primitives from video, and uses game footage to improve terrain-aware humanoid control. The problem is important, and reviewers generally found the paper well motivated, technically solid, and empirically strong.

The overall reviewer consensus was positive, although one reviewer remained unconvinced. The main concerns involved the method's dependence on motion reconstruction quality, the limited scope of the experiments to locomotion and terrain navigation, and whether the notion of "interaction primitives" is fully validated beyond downstream control performance. The rebuttal substantially improved the paper by adding further evidence on robustness, alignment, codebook behaviour, and broader applicability, which resolves many reviewers' technical questions. Nevertheless, Reviewer 1L68 remains a reservation regarding the broader generality of the framework beyond locomotion.

On balance, I find this paper a novel and promising direction for extracting interaction-aware control knowledge from passive video, and the empirical results are strong enough to support publication. I therefore recommend weak acceptance.

This work also went through ethical review regarding the license and copyright conflicts for recording game videos. The authors should acknowledge the legal, non-commercial use of the video data in the revision.